# Quantum Bayesian Optimization

**Zhongxiang Dai\*[1], Gregory Kang Ruey Lau\*[1,2], Arun Verma[1], Yao Shu[1],**
**Bryan Kian Hsiang Low[1], Patrick Jaillet[3]**
[1]Department of Computer Science, National University of Singapore
[2]CNRS@CREATE, 1 Create Way, #08-01 Create Tower, Singapore 138602
[3]Department of Electrical Engineering and Computer Science, MIT
dzx@nus.edu.sg, greglau@comp.nus.edu.sg, arun@comp.nus.edu.sg,
shuyao95@u.nus.edu, lowkh@comp.nus.edu.sg, jaillet@mit.edu

## Abstract

*Kernelized bandits*, also known as *Bayesian optimization* (BO), has been a prevalent method for optimizing complicated black-box reward functions. Various BO algorithms have been theoretically shown to enjoy upper bounds on their *cumulative regret* which are sub-linear in the number $T$ of iterations, and a regret lower bound of $\Omega(\sqrt{T})$ has been derived which represents the unavoidable regrets for any classical BO algorithm. Recent works on *quantum bandits* have shown that with the aid of quantum computing, it is possible to achieve tighter regret upper bounds better than their corresponding classical lower bounds. However, these works are restricted to either multi-armed or linear bandits, and are hence not able to solve sophisticated real-world problems with non-linear reward functions. To this end, we introduce the *quantum-Gaussian process-upper confidence bound* (Q-GP-UCB) algorithm. To the best of our knowledge, our Q-GP-UCB is *the first BO algorithm able to achieve a regret upper bound of $\mathcal{O}(\text{poly}\log T)$*, which is significantly smaller than its regret lower bound of $\Omega(\sqrt{T})$ in the classical setting. Moreover, thanks to our novel analysis of the confidence ellipsoid, our Q-GP-UCB with the linear kernel achieves a smaller regret than the quantum linear UCB algorithm from the previous work. We use simulations, as well as an experiment *using a real quantum computer*, to verify that the theoretical quantum speedup achieved by our Q-GP-UCB is also potentially relevant in practice.

## 1  Introduction

*Kernelized bandits* [9], also named *Bayesian optimization* (BO) [10, 11, 16, 19, 32], has been an immensely popular method for various applications involving the optimization of complicated black-box reward functions. For example, BO has been extensively used to optimize the hyperparameters of machine learning (ML) models [12, 14, 22], the parameters of computationally expensive simulators [5], etc. In every iteration $t = 1, \ldots, T$, BO chooses an arm/input $x_t$ and then queries the reward function $f$ for a noisy observation $y_t = f(x_t) + \zeta_t$ where $\zeta_t$ is a sub-Gaussian noise. In addition to its impressive practical performance, BO is also equipped with solid theoretical guarantees. A number of commonly adopted BO algorithms have been theoretically shown to enjoy sub-linear upper bounds on their *cumulative regret* [9, 33], which ensures that they are guaranteed to be able to find the global optimum of the reward function as $T$ (i.e., the total number of function queries) increases. On the other hand, a lower bound of $\Omega(\sqrt{T})$ (ignoring additional log factors) on the cumulative regret of BO has been shown [31], which represents the fundamental limit of any BO algorithm (in the classical setting). In other words, no classical BO algorithm can achieve a cumulative regret smaller than $\Omega(\sqrt{T})$. This naturally begs the question: *can we leverage more advanced technology to go beyond*

---

\* Equal contribution.

37th Conference on Neural Information Processing Systems (NeurIPS 2023).

*the classical setting and hence break this fundamental limit of* $\Omega(\sqrt{T})$? In this work, we give an affirmative answer by showing that this can be achieved with the aid of *quantum computing* [34].

Quantum bandits, which incorporates quantum computing into bandit algorithms, has been studied by a number of recent works [8, 38, 40, 41]. Notably, the recent work of [38] has introduced quantum variants of classical algorithms for multi-armed bandits (MAB) and stochastic linear bandits (SLB). In the setting of quantum bandits adopted by [38], every query to the reward function $f$ at the arm $x_t$ (in the classical setting) is replaced by a chance to access a *quantum oracle*, which encodes the reward distribution for the arm $x_t$. For every selected arm $x_t$, [38] has proposed to adopt the *quantum Monte Carlo* (QMC) algorithm [4, 28] as a subroutine to obtain an accurate estimation of $f(x_t)$ in an efficient way, i.e., using a small number of queries to the quantum oracle (Lemma 1). As a result, [38] has shown that the regrets of the quantum algorithms for both MAB and SLB are significantly improved compared with their classical counterparts and are smaller than their classical lower bounds. However, both MAB and SLB fall short when optimizing complex real-world functions (e.g., optimizing the hyperparameters of ML models). This is because MAB is unable to exploit the correlations among the arms to model the reward function, and the assumption of a linear reward function adopted by SLB is usually too restrictive in practice. Therefore, designing a quantum bandit algorithm capable of optimizing sophisticated non-linear reward functions, which is a critical step towards practically useful quantum bandit algorithms, is still an open problem. In this work, we resolve this open problem by proposing the first algorithm for quantum BO.

Similar to [38], in our quantum BO problem setting, queries to the reward function in the classical setting are replaced by access to a quantum oracle encoding the reward distribution. As discussed in [38, 39], a quantum oracle is available when the learning environment is implemented by a quantum algorithm, which makes the setting of our quantum BO fairly general. For example, our quantum BO algorithm may be used to optimize the hyperparameters of quantum ML algorithms [3], such as quantum support vector machines (SVMs) [29], quantum neural networks (NNs) [1, 18], among others. It may also be used to optimize the parameters of simulators implemented on a quantum computer, or for applications involving quantum systems where the data produced is inherently quantum. Moreover, our quantum BO algorithm could also be applied to classical data and algorithms, because as discussed in [39], any classical computer program can be converted to a quantum circuit, allowing it to serve as a quantum oracle in our quantum BO algorithm.

In this work, we introduce the first quantum BO algorithm: *quantum-Gaussian process-upper confidence bound* (Q-GP-UCB). For every selected arm $x_s$, our Q-GP-UCB algorithm (Algo. 1) adopts the QMC subroutine (Lemma 1) to efficiently estimate the corresponding reward function value $f(x_s)$ to satisfy a target estimation error $\epsilon_s$. Every arm $x_s$ is selected based on our *weighted* GP posterior distribution, in which every previously selected arm $x_s$ is given a weight of $1/\epsilon_s^2$ which is inversely related to its estimation error. The theoretical analysis of our Q-GP-UCB is faced with non-trivial technical challenges, and our contributions can be summarized as follows:

- We analyze the growth rate of the *weighted information gain* (Sec. 5.1) which arises due to the use of our weighted GP regression (Sec. 4.1), and show its connection with the standard maximum information gain commonly used in the analysis of BO/kernelized bandits [9, 33]. Our analysis here may be of broader independent interest for future works adopting weighted GP regression.
- We derive a tight confidence ellipsoid which gives a guarantee on the concentration of the reward function around the weighted GP posterior mean (Sec. 5.3), and discuss the intuition behind our proof which corresponds to an interesting interpretation about our Q-GP-UCB algorithm. A particularly crucial and novel step in our analysis relies on the recognition to apply the concentration inequality for 1-sub-Gaussian noise (Sec. 5.3).
- We prove that our Q-GP-UCB achieves a regret upper bound of $\mathcal{O}(\text{poly} \log T)$ for the commonly used *squared exponential* (SE) kernel (Secs. 5.4 and 5.5), which is considerably smaller than the classical regret lower bound of $\Omega(\sqrt{T})$ [31] and hence represents a significant quantum speedup.
- By using a similar proof technique to the one adopted for our tight confidence ellipsoid (Sec. 5.3), we improve the confidence ellipsoid for the quantum linear UCB (Q-LinUCB) algorithm [38]. This improved analysis leads to *a tighter regret upper bound for Q-LinUCB*, which matches the regret of our Q-GP-UCB with the linear kernel (Sec. 5.6).
- We use simulations implemented based on the `Qiskit` package to verify the empirical improvement of our Q-GP-UCB over the classical GP-UCB and over Q-LinUCB, through a synthetic experiment and an experiment on *automated machine learning* (AutoML) (Sec. 6). Notably, we have also

performed an experiment *using a real quantum computer*, in which our Q-GP-UCB still achieves a consistent performance advantage (Fig. 4, App. J)

## 2  Related Work

A number of recent works have studied bandit algorithms in the quantum setting. The works of [8] and [40] have focused on the problem of pure exploration in quantum bandits. [25] has studied a different problem setting where the learner aims to maximize some property of an unknown quantum state (i.e., the rewards) by sequentially selecting the observables (i.e., actions). The recent work of [38] has introduced algorithms for quantum multi-armed bandits (MAB) and stochastic linear bandits (SLB), and has shown that by incorporating the QMC subroutine into MAB and SLB, tight regret upper bounds can be achieved which are better than the classical lower bounds. More recently, the works of [23] and [41] have followed similar approaches to introduce quantum bandit algorithms for, respectively, stochastic convex bandits and bandits with heavy-tailed reward distributions. In addition to quantum bandits, some recent works have introduced quantum reinforcement learning (RL) algorithms. [39] has assumed access to a generative model of the environment and proved that their quantum RL algorithm achieves significantly better sample complexity over their classical counterparts. More recently, [17, 43] have removed the requirement for a generative model in quantum RL and achieved quantum speedup in terms of the regret. More comprehensive reviews of the related works on quantum RL can be found in recent surveys (e.g., [27]). The problem setting of our quantum BO (Sec. 3.2) is the same as many of these above-mentioned previous works on quantum bandits [38, 41] and quantum RL [17, 39, 43]. However, to the best of our knowledge, none of the existing works on quantum bandits (and quantum RL) is able to handle non-linear reward functions, which we resolve in this work. Over the years, extensive efforts [7, 24, 30, 36] have been made to design BO/kernelized bandit algorithms whose regret upper bounds are small enough to (nearly) match the classical regret lower bound of $\Omega(\sqrt{T})$ (ignoring additional log factors). Our work provides an alternative route by proving that with the aid of quantum computing, it is possible to achieve a tight regret upper bound which is significantly smaller than the classical regret lower bound.

## 3  Problem Setting and Background

### 3.1  Classical Kernelized Bandits

A BO/kernelized bandit algorithm aims to maximize a reward function $f : \mathcal{X} \to \mathbb{R}$ where $\mathcal{X} \subset \mathbb{R}^d$, i.e., it aims to find $x^* \in \arg\max_{x \in \mathcal{X}} f(x)$. Consistent with the previous works on BO/kernelized bandits [9, 37], we assume that $f$ lies in the *reproducing kernel Hilbert space* (RKHS) associated with a kernel $k$: $f \in H_k(\mathcal{X})$. That is, we assume that $\|f\|_{H_k} \leq B$ for $B > 0$, in which $\|\cdot\|_{H_k}$ denotes the RKHS norm induced by the kernel $k$. Note that unlike previous works on linear bandits [2], our assumption here allows $f$ to be non-linear w.r.t. the input $x$. In this work, we mainly focus on the widely used *squared exponential* (SE) kernel: $k(x, x') \triangleq \exp(-\|x - x'\|_2^2 / (2l^2))$ in which $l$ is the *length scale*. We also extend our results to the Matérn kernel in Sec. 5.7. Without loss of generality, we assume that $k(\cdot, \cdot) \leq 1$. In the following, we use $[t]$ to denote $\{1, \ldots, t\}$ for simplicity.

In every iteration $t \in [T]$ of BO, an input $x_t$ is selected, after which a corresponding noisy observation $y_t = f(x_t) + \zeta_t$ is collected where $\zeta_t$ is a sub-Gaussian noise (e.g., bounded noise or Gaussian noise). The inputs $x_t$'s are sequentially selected by maximizing an *acquisition function*, which is calculated based on the *Gaussian process* (GP) posterior distribution. Specifically, after $t$ iterations, we use the current history $\mathcal{D}_t \triangleq \{(x_1, y_1), \ldots, (x_t, y_t)\}$ to calculate the GP posterior predictive distribution at $x \in \mathcal{X}$: $\mathcal{N}(\mu_t(x), \sigma_t^2(x))$ where

$$\mu_t(x) \triangleq k_t^\top(x)(K_t + \lambda I)^{-1} Y_t, \quad \sigma_t^2(x) \triangleq k(x, x) - k_t^\top(x)(K_t + \lambda I)^{-1} k_t(x), \tag{1}$$

in which $k_t(x) \triangleq [k(x_\tau, x)]_{\tau \in [t]}^\top$ and $Y_t \triangleq [y_{x_\tau}]_{\tau \in [t]}^\top$ are column vectors, $K_t \triangleq [k(x_\tau, x_{\tau'})]_{\tau, \tau' \in [t]}$ is the $t \times t$ covariance matrix, and $\lambda > 1$ is a regularization parameter. Based on (1), a commonly used acquisition function is GP-UCB [33], which selects the next query by: $x_{t+1} = \arg\max_{x \in \mathcal{X}} \mu_t(x) + \xi_{t+1} \sigma_t(x)$ where $\xi_{t+1}$ is a parameter carefully chosen to balance exploration and exploitation. The performance of a BO/kernelized bandit algorithm is usually theoretically analyzed by deriving an upper bound on its cumulative regret: $R_T = \sum_{t=1}^T [f(x^*) - f(x_t)]$, and a tighter regret upper bound is an indicator of a better theoretical convergence.

## 3.2 Quantum Bandits/BO

A *quantum state* $|x\rangle$ in Hilbert space $\mathbb{C}^n$ can be expressed as a superposition of $n$ basis states (e.g. qubits) via a vector $\vec{x} = [x_1, \ldots, x_n]^\top$, with $|x_i|^2$ representing the probability of being in the $i^{th}$ basis state. Given two quantum states $|x\rangle \in \mathbb{C}^n$ and $|y\rangle \in \mathbb{C}^m$, we use $|x\rangle|y\rangle = [x_1 y_1, \ldots, x_n y_m]^\top$ to denote their tensor product. A quantum algorithm typically works by applying unitary operators to quantum states, and may have access to input data encoded in unitary operators called *quantum oracles* (examples have been discussed in Sec. 1). We defer a more detailed introduction to quantum computing to related works dedicated to these topics (e.g., [26]).

Our quantum BO setting follows that of the quantum bandits works in [38, 41]. In every iteration of quantum BO, after an input $x$ is selected, instead of observing a noisy reward as in classical BO/bandits (Sec. 3.1), we get a chance to access a quantum unitary oracle $\mathcal{O}_x$ and its inverse that encode the noisy reward distribution. Specifically, let $P_x$ denote the reward distribution, $\Omega_x$ denote the finite sample space of $P_x$, and $y_x : \Omega_x \to \mathbb{R}$ denote the random reward associated with input $x$. Then, $\mathcal{O}_x$ is formally defined as:

$$\mathcal{O}_x : |0\rangle \to \sum_{\omega \in \Omega_x} \sqrt{P_x(\omega)} |\omega\rangle |y_x(\omega)\rangle. \tag{2}$$

*Quantum mean estimation*, which aims to estimate the mean of an unknown distribution with better sample efficiency than classical algorithms, has been a widely studied problem [4, 21]. Consistent with the works of [38, 41], we will make use of the following *quantum Monte Carlo* (QMC) algorithm:

**Lemma 1** (Quantum Monte Carlo (QMC) [28]). *Let $y_x : \Omega_x \to \mathbb{R}$ denote a random variable, $\Omega_x$ is equipped with probability measure $P_x$, and the quantum unitary oracle $\mathcal{O}_x$ encodes $P_x$ and $y_x$.*

- ***Bounded Noise**: If the noisy output observation satisfies $y_x \in [0, 1]$, then there exists a constant $C_1 > 1$ and a QMC algorithm $QMC(\mathcal{O}_x, \epsilon, \delta)$ which returns an estimate $\hat{y}_x$ of $\mathbb{E}[y_x]$ such that $\mathbb{P}(|\hat{y}_x - \mathbb{E}[y_x]| > \epsilon) \leq \delta$, using at most $\frac{C_1}{\epsilon} \log(1/\delta)$ queries to $\mathcal{O}_x$ and its inverse.*
- ***Noise with Bounded Variance**: If the variance of $y_x$ is $\leq \sigma^2$, then for $\epsilon < 4\sigma$, there exists a constant $C_2 > 1$ and a QMC algorithm $QMC(\mathcal{O}_x, \epsilon, \delta)$ which returns an estimate $\hat{y}_x$ s.t. $\mathbb{P}(|\hat{y}_x - \mathbb{E}[y_x]| > \epsilon) \leq \delta$, using at most $\frac{C_2 \sigma}{\epsilon} \log_2^{3/2}\left(\frac{8\sigma}{\epsilon}\right) \log_2(\log_2 \frac{8\sigma}{\epsilon}) \log \frac{1}{\delta}$ queries to $\mathcal{O}_x$ and its inverse.*

The *quantum query complexity* of a quantum algorithm is usually measured by *the number of queries to the quantum oracle* [3, 38]. So, the sample complexities from Lemma 1 can be compared with that of classical Monte Carlo (MC) estimation. In the classical setting, it can be easily shown using concentration inequalities that MC estimation requires $\widetilde{\mathcal{O}}(1/\epsilon^2)$ samples to reach a target mean estimation error of $\epsilon$. Therefore, the QMC algorithm (Lemma 1), which only needs $\widetilde{\mathcal{O}}(1/\epsilon)$ samples, achieves a quadratic reduction in the required number of samples. This dramatic improvement is crucial for the quantum speedup in terms of regrets achieved by our algorithm (Sec. 5.4).

During our Q-GP-UCB algorithm, after every input $x_s$ is selected, we will apply the QMC algorithm from Lemma 1 with the quantum oracle $\mathcal{O}_{x_s}$ to obtain an estimation $y_s$ of its reward $f(x_s)$ (line 6 of Algo. 1). Of note, the above-mentioned equivalence between a query to the quantum oracle and a sample for classical MC mean estimation implies that *a query to the quantum oracle $\mathcal{O}_x$ in quantum BO/bandits is equivalent to the pulling of an arm $x$ in classical BO/bandits*. Therefore, the cumulative regret $R_T$ of our Q-GP-UCB algorithm is defined as the regret incurred after $T$ queries to the quantum oracle, which makes it amenable to comparisons with the regrets of classical algorithms.

## 4 Quantum Bayesian Optimization

In this section, we first introduce weighted GP regression (Sec. 4.1) which will be used by our Q-GP-UCB algorithm to calculate the acquisition function for input selection, and then describe our Q-GP-UCB algorithm (Sec. 4.2) in detail.

### 4.1 Weighted GP Posterior Distribution

In contrast to standard GP-UCB [33] which uses the standard GP posterior distribution (1) to calculate the acquisition function, our Q-GP-UCB makes use of a *weighted* GP posterior distribution. That is, we assign a weight to every previous observation. Specifically, after stage $s$ of our Q-GP-UCB (i.e., given $\mathcal{D}_s \triangleq \{(x_\tau, y_\tau)\}_{\tau \in [s]}$), we define the weight matrix $W_s \triangleq \text{diag}(1/\epsilon_1^2, \ldots, 1/\epsilon_s^2)$. $W_s$ is

an $s \times s$ diagonal matrix, in which the $\tau^{\text{th}}$ diagonal element represents the weight $1/\epsilon_\tau^2$ given to the $\tau^{\text{th}}$ observation $(x_\tau, y_\tau)$. We will set $\epsilon_\tau = \widetilde{\sigma}_{\tau-1}(x_\tau)/\sqrt{\lambda}$ (Sec. 4.2), i.e., $\epsilon_\tau$ is calculated using the weighted GP posterior standard deviation (3) (conditioned on the first $\tau - 1$ observations) at $x_\tau$.

Define $K_s \triangleq [k(x_\tau, x_{\tau'})]_{\tau,\tau'\in[s]}$ which is the $s \times s$-dimensional covariance matrix given the first $s$ observations, and define $\widetilde{K}_s \triangleq W_s^{1/2} K_s W_s^{1/2}$ which is the *weighted* covariance matrix. Similarly, define $k_s(x) \triangleq [k(x, x_\tau)]_{\tau\in[s]}^\top$ and $\widetilde{k}_s(x) \triangleq W_s^{1/2} k_s(x)$. Denote the collection of output observations by $Y_s \triangleq [y_\tau]_{\tau\in[s]}^\top$, and define $\widetilde{Y}_s \triangleq W_s^{1/2} Y_s$. With these definitions, given $\mathcal{D}_s$, our weighted GP posterior distribution at an input $x \in \mathcal{X}$ is a Gaussian distribution: $\mathcal{N}(\widetilde{\mu}_s(x), \widetilde{\sigma}_s^2(x))$, in which

$$\widetilde{\mu}_s(x) \triangleq \widetilde{k}_s^\top(x)(\widetilde{K}_s + \lambda I)^{-1}\widetilde{Y}_s, \quad \widetilde{\sigma}_s^2(x) \triangleq k(x,x) - \widetilde{k}_s^\top(x)(\widetilde{K}_s + \lambda I)^{-1}\widetilde{k}_s(x). \quad (3)$$

Note that the GP posterior mean $\widetilde{\mu}_s$ above is equivalently the solution to the following weighted kernel ridge regression problem: $\widetilde{\mu}_s = \arg\min_{f \in H_k(\mathcal{X})} \sum_{\tau=1}^s \frac{1}{\epsilon_\tau^2}(y_\tau - f(x_\tau))^2 + \lambda\|f\|_{H_k}^2$. We give a more detailed analysis of the weighted GP posterior (3) in App. A. Weighted GP regression has also been adopted by previous works on BO such as [15]. However, our choice of the weights, algorithmic design and theoretical analyses all require significantly novel treatments.

## 4.2 Q-GP-UCB Algorithm

---
**Algorithm 1** Q-GP-UCB

---
1: **for** stage $s = 1, 2, \ldots$ **do**
2: $\quad x_s = \arg\max_{x \in \mathcal{X}} \widetilde{\mu}_{s-1}(x) + \beta_s \widetilde{\sigma}_{s-1}(x)$ (3).
3: $\quad \epsilon_s = \widetilde{\sigma}_{s-1}(x_s)/\sqrt{\lambda}$.
4: $\quad$ *(a)* $N_{\epsilon_s} = \frac{C_1}{\epsilon_s} \log(\frac{2\overline{m}}{\delta})$ (for bounded noise), or
$\quad\quad$ *(b)* $N_{\epsilon_s} = \frac{C_2\sigma}{\epsilon_s} \log_2^{3/2}\left(\frac{8\sigma}{\epsilon_s}\right) \log_2(\log_2 \frac{8\sigma}{\epsilon_s}) \log \frac{2\overline{m}}{\delta}$ (for noise with variance bounded by $\sigma^2$).
5: $\quad$ If $\sum_{k=1}^s N_{\epsilon_k} > T$, terminate the algorithm.
6: $\quad$ Run the QMC$(\mathcal{O}_{x_s}, \epsilon_s, \delta/(2\overline{m}))$ algorithm to query the quantum oracle of $x_s$ for the next $N_{\epsilon_s}$ rounds, and obtain $y_s$ as an estimate of $f(x_s)$.
7: $\quad$ Update the weighted GP posterior (3) using $(x_s, y_s)$.

---

Our Q-GP-UCB algorithm is presented in Algo. 1. Q-GP-UCB proceeds in stages. In stage $s$, we first select the next input $x_s$ to query by maximizing the GP-UCB acquisition function calculated using the weighted GP posterior distribution (3) (line 2 of Algo. 1). Here $\beta_s \triangleq B + \sqrt{2(\widetilde{\gamma}_{s-1} + 1 + \log(2/\delta))}$ (more details in Sec. 5.3), in which $\delta \in (0, 2/e]$ and $\widetilde{\gamma}_{s-1} \triangleq \frac{1}{2}\log(\det(I + \frac{1}{\lambda}\widetilde{K}_{s-1}))$ is the *weighted information gain* (more details in Sec. 5.1). Next, we calculate $\epsilon_s = \widetilde{\sigma}_{s-1}(x_s)/\sqrt{\lambda}$ (line 3) and $N_{\epsilon_s}$ (line 4), in which $N_{\epsilon_s}$ depends on the type of noise and the value of $\epsilon_s$. Here $\overline{m}$ is an upper bound on the total number $m$ of stages which we will analyze in Sec. 5.2. Subsequently, unless the algorithm is terminated (line 5), we run the QMC algorithm (Lemma 1) to estimate $f(x_s)$ by querying the quantum oracle of $x_s$ for $N_{\epsilon_s}$ rounds (line 6). The QMC procedure returns an estimate $y_s$ of the reward function value $f(x_s)$, for which the estimation error is guaranteed to be bounded: $|y_s - f(x_s)| \leq \epsilon_s$ with probability of at least $1 - \delta/(2\overline{m})$. Lastly, we update the weighted GP posterior (3) using the newly collected input-output pair $(x_s, y_s)$, as well as its weight $1/\epsilon_s^2$ (line 7).

Of note, the value of $\epsilon_s$ is used for both **(a)** calculating the number $N_{\epsilon_s}$ of queries to the quantum oracle for $x_s$, and **(b)** computing the weight $1/\epsilon_s^2$ assigned to $(x_s, y_s)$ in the weighted GP regression (3) in the subsequent iterations. Regarding **(a)**, our designs of $\epsilon_s$ and $N_{\epsilon_s} = \widetilde{\mathcal{O}}(1/\epsilon_s)$ have an interesting interpretation in terms of the exploration-exploitation trade-off: In the initial stages, the weighted GP posterior standard deviation $\widetilde{\sigma}_{s-1}(x_s)$ and hence $\epsilon_s$ are usually large, which leads to a small number $N_{\epsilon_s}$ of queries for every $x_s$ and hence allows our Q-GP-UCB to favor the *exploration* of more unique inputs; in later stages, $\widetilde{\sigma}_{s-1}(x_s)$ and $\epsilon_s$ usually become smaller, which results in large $N_{\epsilon_s}$'s and hence causes our Q-GP-UCB to prefer the *exploitation* of a small number of unique inputs. Regarding **(b)**, assigning a larger weight $1/\epsilon_s^2$ to an input $x_s$ with a smaller $\epsilon_s$ is reasonable, because a smaller $\epsilon_s$ indicates a smaller estimation error for $y_s$ as we explained above, which makes the observation $y_s$ more accurate and reliable for calculating the weighted GP regression (3).

Note that we have modified the original GP-UCB by querying every selected input $x_s$ multiple times, in order to make it amenable to the integration of the QMC subroutine. The recent work of [6] has also

adapted BO to evaluate a small number of unique inputs while querying each of them multiple times. It has been shown [6] that the resulting BO algorithm preserves both the theoretical guarantee (i.e., the regret upper bound) and empirical performance of the original BO, while significantly reducing the computational cost. So, the findings from [6] can serve as justifications for our modification to GP-UCB. Also note that despite this similarity, [6] still focuses on the traditional setting (i.e., their regret upper bound is $R_T = \mathcal{O}(\gamma_T \sqrt{T})$) and their regret analyses are entirely different from ours.

## 5 Theoretical Analysis

Throughout our analysis, we condition on the event that $|f(x_s) - y_s| \leq \epsilon_s, \forall s = 1, \ldots, m$. This event holds with probability of at least $1 - \delta/2$, because we set the error probability in QMC (Lemma 1) to $\delta/(2\overline{m})$ in which $\overline{m} \geq m$ (see Theorem 2 for details). For simplicity, we mainly focus on the scenario of bounded noise (i.e., the observations are bounded within $[0, 1]$, the first case of Lemma 1), because the theoretical analyses and insights about the other scenario of noise with bounded variance (i.e., the second case of Lemma 1) are almost identical (more details in Sec. 5.5).

### 5.1 Weighted Information Gain

To begin with, the following lemma (proof in App. B) gives an upper bound on the sum of the weights in all $m$ stages, which will be useful for upper-bounding the weighted information gain $\widetilde{\gamma}_m$.

**Lemma 2.** *Choose $\delta \in (0, 2/e]$, then we have that $\sum_{s=1}^{m} 1/\epsilon_s^2 \leq T^2$.*

Denote by $\gamma_T$ the maximum information gain from any set of $T$ inputs: $\gamma_T \triangleq \max_{\{x_1, \ldots, x_T\} \subset \mathcal{X}} \frac{1}{2} \log \det(I + \frac{1}{\lambda} K_T)$, which is commonly used in the analysis of BO/kernelized bandit algorithms [9, 33]. Denote by $\widetilde{\gamma}_m$ the *weighted information gain* from all $m$ selected inputs in all stages: $\widetilde{\gamma}_m \triangleq \frac{1}{2} \log \det(I + \frac{1}{\lambda} \widetilde{K}_m)$. Note that in the definition of $\widetilde{\gamma}_m$, we do not take the maximum over all sets of $m$ inputs, so $\widetilde{\gamma}_m$ still depends on the selected inputs $\mathcal{X}_m \triangleq \{x_1, \ldots, x_m\}$. Also note that $\{\epsilon_1, \ldots, \epsilon_m\}$ are known constants conditioned on $\mathcal{X}_m$. This is because the weighted GP posterior standard deviation $\widetilde{\sigma}_{s-1}$ (3) does not depend on the output observations, so every $\epsilon_s = \widetilde{\sigma}_{s-1}(x_s)/\sqrt{\lambda}$ only depends on $\{x_1, \ldots, x_s\}$. The next theorem gives an upper bound on $\widetilde{\gamma}_m$.

**Theorem 1.** *(a) Given $\mathcal{X}_m \triangleq \{x_1, \ldots, x_m\}$, the growth rate of $\widetilde{\gamma}_m$ is the same as that of $\gamma_{\sum_{s=1}^m 1/\epsilon_s^2}$. (b) $\widetilde{\gamma}_m \leq \gamma_{T^2}$.*

We give the complete statement of result *(a)* in Theorem 8 (App. C), which presents the concrete growth rate. As stated by result *(a)*, to obtain an upper bound on $\widetilde{\gamma}_m$, we simply need to firstly use the upper bound on $\gamma_T$ from previous works [35], and then replace the $T$ in this upper bound by $\sum_{s=1}^m 1/\epsilon_s^2$. Based on this, the result *(b)* has further upper-bounded $\sum_{s=1}^m 1/\epsilon_s^2$ by $T^2$ through Lemma 2. Note that because $\epsilon_s = \widetilde{\sigma}_{s-1}(x_s)/\sqrt{\lambda} \leq 1$, we have that $1/\epsilon_s^2 \geq 1$ and hence $\sum_{s=1}^m 1/\epsilon_s^2 \geq m$. Therefore, our weighted information gain $\widetilde{\gamma}_m$ has a looser upper bound than $\gamma_m$. This intuitively implies that our weighted covariance matrix $\widetilde{K}_m$ is expected to *contain more information than the original $K_m$*, whose implication will be discussed again in Sec. 5.2. The proof of Theorem 1 (App. C) has followed the analysis of the information gain from [35]. Specifically, we have leveraged a finite-dimensional projection of the infinite-dimensional RKHS feature space, and hence upper-bounded the information gain in terms of the tail mass of the eigenvalues of the kernel $k$. The proof requires carefully tracking the impact of the weights $1/\epsilon_s^2$ throughout the analysis. Importantly, our Theorem 1 and its analysis may be *of wider independent interest* for future works which also adopt weighted GP regression (Sec. 4.1).

The growth rate of $\gamma_T$ has been characterized for commonly used kernels [35]. Based on these, the next corollary provides the growth rates of $\widetilde{\gamma}_m$ for the linear and squared exponential (SE) kernels.

**Corollary 1.** *For the linear and squared exponential (SE) kernels, we have that*

$$\widetilde{\gamma}_m = \mathcal{O}(d \log \sum_{s=1}^m 1/\epsilon_s^2) = \mathcal{O}(d \log T) \qquad \textit{linear kernel,}$$
$$\widetilde{\gamma}_m = \mathcal{O}(\log^{d+1} \sum_{s=1}^m 1/\epsilon_s^2) = \mathcal{O}((\log T)^{d+1}) \qquad \textit{SE kernel.}$$

### 5.2 Upper Bound on The Total Number $m$ of Stages

The next theorem gives an upper bound on the total number of stages of our Q-GP-UCB algorithm.

**Theorem 2.** *For the linear kernel, our Q-GP-UCB algorithm has at most $m = \mathcal{O}(d \log T)$ stages. For the SE kernel, our Q-GP-UCB algorithm has at most $m = \mathcal{O}((\log T)^{d+1})$ stages.*

Note that for the linear kernel, our upper bound on $m$ matches that of the Q-LinUCB algorithm from [38] (see Lemma 2 from [38]). The proof of Theorem 2 is given in App. D, and here we give a brief sketch of the proof. For stage $s$, define $V_s \triangleq \lambda I + \sum_{\tau=1}^{s}(1/\epsilon_\tau^2)\phi(x_\tau)\phi(x_\tau)^\top$, in which $\phi(\cdot)$ is the (potentially infinite-dimensional) RKHS feature mapping: $k(x, x') = \phi(x)^\top\phi(x')$. Intuitively, $\log \det V_s$ represents the amount of information contained in the first $s$ selected inputs $\{x_1, \ldots, x_s\}$. As the first step in our proof, we prove that thanks to our choice of $\epsilon_s$ (line 3 of Algo. 1) and our design of the weights $1/\epsilon_s^2$, the amount of information $\log \det V_s$ is doubled after every stage: $\det V_{\tau+1} = 2 \det V_\tau, \forall \tau = 0, \ldots, m-1$. This allows us to show that $m \log 2 = \log(\det(V_m)/\det(V_0))$ where $V_0 = \lambda I$. Next, we show that this term is also related to our weighted information gain: $\log(\det(V_m)/\det(V_0)) = 2\widetilde{\gamma}_m$, which allows us to upper-bound $m$ in terms of $\widetilde{\gamma}_m$ and hence in terms of $\sum_{s=1}^{m} 1/\epsilon_s^2$ (see Corollary 1). This eventually allows us to derive an upper bound on $m$ by using the fact that the total number of iterations (i.e., queries to the quantum oracles) is no larger than $T$. Therefore, the key intuition of the proof here is that as long as we gather a sufficient amount of information in every stage, we do not need a large total number of stages.

### 5.3  Confidence Ellipsoid

The next theorem proves the concentration of $f$ around the weighted GP posterior mean (3).

**Theorem 3.** *Let $\eta \triangleq 2/T$, $\lambda \triangleq 1 + \eta$, and $\beta_s \triangleq B + \sqrt{2(\widetilde{\gamma}_{s-1} + 1 + \log(2/\delta))}$. We have with probability of at least $1 - \delta/2$ that*

$$|f(x) - \widetilde{\mu}_{s-1}(x)| \leq \beta_s \widetilde{\sigma}_{s-1}(x), \qquad \forall s \in [m], x \in \mathcal{X}.$$

The proof of Theorem 3 is presented in App. E, and here we give a brief proof sketch. Denote by $\mathcal{F}_{s-1}$ the $\sigma$-algebra $\mathcal{F}_{s-1} = \{x_1, \ldots, x_s, \zeta_1, \ldots, \zeta_{s-1}\}$ where $\zeta_k = y_k - f(x_k)$ is the noise. Recall that $W_s^{1/2} \triangleq \operatorname{diag}(1/\epsilon_1, \ldots, 1/\epsilon_s)$ (Sec. 4.1), and define $\zeta_{1:s} \triangleq [\zeta_k]_{k \in [s]}$. In the first part of the proof, we use the RKHS feature space view of the weighted GP posterior (3) (see App. A) to upper-bound $|f(x) - \widetilde{\mu}_{s-1}(x)|$ in terms of $\widetilde{\sigma}_{s-1}(x)$ and $\|W_s^{1/2}\zeta_{1:s}\|_{((\widetilde{K}_s + \eta I)^{-1} + I)^{-1}}$. Next, the most crucial step in the proof is to upper-bound $\|W_s^{1/2}\zeta_{1:s}\|_{((\widetilde{K}_s + \eta I)^{-1} + I)^{-1}}$ in terms of $\widetilde{\gamma}_s$ by applying the self-normalized concentration inequality from Theorem 1 of [9] to *1-sub-Gaussian noise*. This is feasible because *(a)* every $\epsilon_s = \widetilde{\sigma}_{s-1}(x_s)/\sqrt{\lambda}$ is $\mathcal{F}_{s-1}$-measurable, and *(b)* conditioned on $\mathcal{F}_{s-1}$, the scaled noise term $\zeta_s/\epsilon_s$ (in $W_s^{1/2}\zeta_{1:s}$) is 1-sub-Gaussian. The statement *(a)* follows because $\epsilon_s$ only depends on $\{x_1, \ldots, x_s\}$ as we have discussed in Sec. 5.1. The statement *(b)* follows since our QMC subroutine ensures that every noise term $\zeta_k$ is bounded within $[-\epsilon_k, \epsilon_k]$ (Sec. 4.2) because $|y_k - f(x_k)| \leq \epsilon_k$ (with high probability). This suggests that conditioned on $\mathcal{F}_{s-1}$ (which makes $\epsilon_s$ a known constant as discussed above), $\zeta_s/\epsilon_s$ is bounded within $[-1, 1]$ and is hence 1-sub-Gaussian.

This critical step in the proof is in fact also consistent with an interesting interpretation about our algorithm. That is, after $x_s$ is selected in stage $s$, we adaptively decide the number $N_{\epsilon_s}$ of queries to the quantum oracle in order to reach the target accuracy $\epsilon_s$ (line 4 of Algo. 1). This ensures that $|y_s - f(x_s)| \leq \epsilon_s$ (with high probability), which guarantees that the noise $\zeta_s = y_s - f(x_s)$ is (conditionally) $\epsilon_s$-sub-Gaussian. Moreover, note that the vector of observations $\widetilde{Y}_s$ used in the calculation of the weighted GP posterior mean (3) is $\widetilde{Y}_s = W_s^{1/2}Y_s$. So, from the perspective of the weighted GP posterior (3), every noisy observation $y_s$, and hence every noise $\zeta_s$, is multiplied by $1/\epsilon_s$. So, the effective noise of the observations $\widetilde{Y}_s$ used in the weighted GP posterior (i.e., $\zeta_s/\epsilon_s$) is $\epsilon_s/\epsilon_s = 1$-sub-Gaussian. This shows the underlying intuition as to why our proof of the concentration of the reward function $f$ using the weighted GP regression can make use of *1-sub-Gaussian noise* (e.g., in contrast to the $\sigma$-sub-Gaussian noise used in the proof of classical GP-UCB [9]).

Our tight confidence ellipsoid around $f$ from Theorem 3 is crucial for deriving a tight regret upper bound for our Q-GP-UCB (Sec. 5.4). Moreover, as we will discuss in more detail in Sec. 5.6, our proof technique for Theorem 3, interestingly, can be adopted to obtain a tighter confidence ellipsoid for the Q-LinUCB algorithm from [38] and hence improve its regret upper bound.

### 5.4  Regret Upper Bound

Here we derive an upper bound on the cumulative regret of our Q-GP-UCB algorithm.

**Theorem 4.** *With probability of at least* $1 - \delta$*, we have that*[1]

$$R_T = \mathcal{O}\big(d^{3/2}(\log T)^{3/2}\log(d\log T)\big) \qquad linear\ kernel,$$
$$R_T = \mathcal{O}\big((\log T)^{3(d+1)/2}\log((\log T)^{d+1})\big) \qquad SE\ kernel.$$

The proof of Theorem 4 is presented in App. F. The proof starts by following the standard technique in the proof of UCB-type algorithms to show that $r_s = f(x^*) - f(x_s) \leq 2\beta_s\widetilde{\sigma}_{s-1}(x_s)$. Next, importantly, our design of $\epsilon_s = \widetilde{\sigma}_{s-1}(x_s)/\sqrt{\lambda}$ allows us to show that $r_s \leq 2\beta_s\epsilon_s\sqrt{\lambda}$. After that, the total regrets incurred in a stage $s$ can be upper-bounded by $N_{\epsilon_s} \times 2\beta_s\epsilon_s\sqrt{\lambda} = \mathcal{O}(\frac{1}{\epsilon_s}\log(\overline{m}/\delta) \times \beta_s\epsilon_s) = \mathcal{O}(\beta_s\log(\overline{m}/\delta))$. Lastly, the regret upper bound follows by summing the regrets from all $m$ stages. Of note, for the commonly used SE kernel, our regret upper bound is of the order $\mathcal{O}(\text{poly}\log T)$, which is significantly smaller than the regret lower bound of $\Omega(\sqrt{T})$ in the classical setting shown by [31]. This improvement over the classical fundamental limit is mostly attributed to the use of the QMC (Lemma 1), i.e., line 6 of Algo. 1. If the QMC procedure is replaced by classical MC estimation, then in contrast to the $N_{\epsilon_s} = \mathcal{O}(1/\epsilon_s)$ queries by QMC, every stage would require $N_{\epsilon_s} = \mathcal{O}(1/\epsilon_s^2)$ queries. As a result, this would introduce an additional factor of $1/\epsilon_s$ to the total regrets in a stage $s$ (as discussed above), which would render the derivation of our regret upper bound (Theorem 4) invalid. This verifies that our tight regret upper bound in Theorem 4 is only achievable with the aid of quantum computing. Our Theorem 4 is a demonstration of the immense potential of quantum computing to dramatically improve the theoretical guarantees over the classical setting.

Importantly, when the linear kernel is used, the regret upper bound of our Q-GP-UCB is in fact better than that of the Q-LinUCB algorithm from [38]. This improvement arises from our tight confidence ellipsoid in Theorem 3. In Sec. 5.6, we will give a more detailed discussion of this improvement and adopt our proof technique for Theorem 3 to improve the confidence ellipsoid for Q-LinUCB, after which their improved regret upper bound matches that of our Q-GP-UCB with the linear kernel.

### 5.5 Regret Upper Bound for Noise with Bounded Variance

Here we analyze the regret of our Q-GP-UCB when the variance of the noise is bounded by $\sigma^2$, which encompasses common scenarios including $\sigma$-sub-Gaussian noises and Gaussian noises with a variance of $\sigma^2$. Consistent with the analysis of Q-LinUCB [38], in order to apply the theoretical guarantee provided by QMC (Lemma 1) for noise with bounded variance, here we need to assume that the noise variance is not too small, i.e., we assume that $\sigma > 1/4$. In this case of noise with bounded variance, the following theorem gives an upper bound on the regret of our Q-GP-UCB.

**Theorem 5.** *With probability of at least* $1 - \delta$*, we have that*

$$R_T = \mathcal{O}\big(\sigma d^{3/2}(\log T)^{3/2}\log(d\log T)(\log_2\sigma T)^{3/2}\log_2(\log_2\sigma T)\big) \qquad linear\ kernel,$$
$$R_T = \mathcal{O}\big(\sigma(\log T)^{3(d+1)/2}\log((\log T)^{d+1})(\log_2\sigma T)^{3/2}\log_2(\log_2\sigma T)\big) \qquad SE\ kernel. \tag{4}$$

The proof of Theorem 5 is given in App. G. Note that same as Theorem 4, the regret for the SE kernel in Theorem 5 is also of the order $\mathcal{O}(\text{poly}\log T)$, which is also significantly smaller than the classical regret lower bound of $\Omega(\sqrt{T})$ [31]. Moreover, Theorem 5 shows that for both the linear kernel and SE kernel, the regret of our Q-GP-UCB is reduced when the noise variance $\sigma^2$ becomes smaller.

### 5.6 Improvement to Quantum Linear UCB (Q-LinUCB)

As we have mentioned in Sec. 5.4, the regret upper bound of our Q-GP-UCB with the linear kernel is $R_T = \mathcal{O}(d^{3/2}(\log T)^{3/2}\log(d\log T))$, which is better than the corresponding $R_T = \mathcal{O}(d^2(\log T)^{3/2}\log(d\log T))$ for Q-LinUCB [38][2] by a factor of $\mathcal{O}(\sqrt{d})$. This improvement can be attributed to our tight confidence ellipsoid from Theorem 3. Here, following the general idea of the proof of our Theorem 3, we prove a tighter confidence ellipsoid for the Q-LinUCB algorithm, i.e., we improve Lemma 3 from the work of [38]. Here we follow the notations from [38], and we defer the detailed notations, as well as the proof of Theorem 6 below, to App. H due to space limitation.

---

[1]We have omitted the dependency on $\log(1/\delta)$ for simplicity. Refer to App. F for the full expression.

[2]This is the *high-probability* regret (rather than expected regret) from [38], which we focus on in this work.

**Theorem 6** (Improved Confidence Ellipsoid for [38]). *With probability of at least $1 - \delta$,*

$$\theta^* \in \mathcal{C}_s = \{\theta \in \mathbb{R}^d : \left\|\theta - \widehat{\theta}_s\right\|_{V_s} \leq \Big(\sqrt{2d \log\Big(1 + \frac{L^2}{\lambda}\sum_{k=1}^{s}\frac{1}{\epsilon_k^2}\Big)\frac{1}{\delta}} + \sqrt{\lambda}S\Big)\}, \qquad \forall s \in [m].$$

Note that the size of the confidence ellipsoid from our Theorem 6 is of the order $\mathcal{O}(\sqrt{d\log(\sum_{k=1}^{s}1/\epsilon_k^2)}) = \mathcal{O}(\sqrt{d\log T})$ (we have used Lemma 2 here), which improves over the $\mathcal{O}(\sqrt{ds}) = \mathcal{O}(\sqrt{dm}) = \mathcal{O}(\sqrt{d \times d\log T}) = \mathcal{O}(d\sqrt{\log T})$ from Lemma 3 of [38] by a factor of $\sqrt{d}$. Plugging in our tighter confidence ellipsoid (Theorem 6) into the regret analysis of Q-LinUCB, the regret upper bound for Q-LinUCB is improved to $R_T = \mathcal{O}(d^{3/2}\log^{3/2}T\log(d\log T))$ (more details at the end of App. H). This improved regret upper bound exactly matches that of our Q-GP-UCB with the linear kernel (Theorem 4). Similar to Theorem 3, the most crucial step in the proof of Theorem 6 is to apply the self-normalized concentration inequality from Theorem 1 of [2] to *1-sub-Gaussian noise*. Again this is feasible because $\epsilon_s = \widetilde{\sigma}_{s-1}(x_s)/\sqrt{\lambda}$ is $\mathcal{F}_{s-1}$-measurable, and conditioned on $\mathcal{F}_{s-1}$, the scaled noise $\zeta_s/\epsilon_s$ is 1-sub-Gaussian.

### 5.7 Regret Upper Bound for the Matérn Kernel

So far we have mainly focused on the linear and SE kernels in our analysis. Here we derive a regret upper bound for our Q-GP-UCB algorithm for the Matérn kernel with smoothness parameter $\nu$.

**Theorem 7** (Matérn Kernel, Bounded Noise). *With probability of at least $1 - \delta$, we have that*

$$R_T = \widetilde{\mathcal{O}}\big(T^{3d/(2\nu+d)}\big).$$

The proof is in App. I. Note that the state-of-the-art regret upper bound for the Matérn kernel in the classical setting is $R_T = \mathcal{O}(\sqrt{T\gamma_T}) = \widetilde{\mathcal{O}}(T^{(\nu+d)/(2\nu+d)})$ [7, 24, 30, 36], which matches the corresponding classical regret lower bound (up to logarithmic factors in $T$) [35]. Therefore, the regret upper bound of our Q-GP-UCB for the Matérn kernel (Theorem 7) improves over the corresponding classical regret lower bound when $\nu > 2d$, i.e., when the reward function is sufficiently smooth. Also note that similar to the classical GP-UCB [33] (with regret $R_T = \mathcal{O}(\gamma_T\sqrt{T}) = \widetilde{\mathcal{O}}(T^{(\nu+3d/2)/(2\nu+d)})$) which requires the reward function to be sufficiently smooth (i.e., $\nu > d/2$) to attain sub-linear regrets for the Matérn kernel, our Q-GP-UCB, as the first quantum BO algorithm, also requires the reward function to be smooth enough (i.e., $\nu > d$) in order to achieve a sub-linear regret upper bound for the Matérn kernel. We leave it to future works to further improve our regret upper bound and hence relax this requirement for smooth functions.

## 6 Experiments

We use the `Qiskit` python package to implement the QMC algorithm (Lemma 1) following the recent work of [20]. Some experimental details are deferred to App. J due to space limitation.

**Synthetic Experiment.** Here we use a grid of $|\mathcal{X}| = 20$ equally spaced points within $[0, 1]$ as the 1-dimensional input domain $\mathcal{X}$ ($d = 1$), and sample a function $f$ from a GP prior with the SE kernel. The sampled function $f$ is scaled so that its output is bounded within $[0, 1]$, and then used as the reward function in the synthetic experiments. We consider two types of noises: *(a)* bounded noise (within $[0, 1]$) and *(b)* Gaussian noise, which correspond to the two types of noises in Lemma 1, respectively. For *(a)* bounded noise, we follow the practice of [38] such that when an input $x$ is selected, we treat the function value $f(x)$ as the probability for a Bernoulli distribution, i.e., we observe an output of 1 with probability of $f(x)$ and 0 otherwise. For *(b)* Gaussian noise, we simply add a zero-mean Gaussian noise with variance $\sigma^2$ to $f(x)$. The results for *(a)* bounded noise and *(b)* Gaussian noise are shown in Figs. 1 (a) and (b), respectively. The figures show that for both types of noises, our Q-GP-UCB significantly outperforms the classical baseline of GP-UCB. Specifically, although our Q-GP-UCB incurs larger regrets in the initial stage, it is able to leverage the accurate observations provided by the QMC subroutine to rapidly find the global optimum. These results show that the quantum speedup of our Q-GP-UCB in terms of the tighter regret upper bounds (Theorems 4 and 5) may also be relevant in practice. We have additionally compared with linear UCB (LinUCB) [2] and Q-LinUCB [38], which are, respectively, the most representative classical and quantum linear bandit algorithms. The results in Fig. 2 (App. J) show that in these experiments where the reward

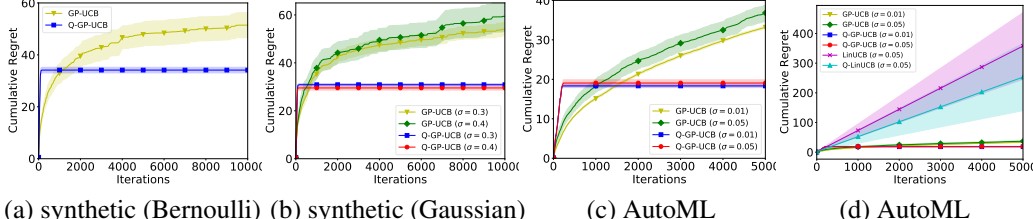

(a) synthetic (Bernoulli)  (b) synthetic (Gaussian)  (c) AutoML  (d) AutoML

Figure 1: Cumulative regret for synthetic experiments with (a) bounded noise and (b) Gaussian noise. (c) Cumulative regret for the AutoML experiment; (d) additionally includes results for linear bandits.

function is non-linear, algorithms based on linear bandits severely underperform compared with BO/kernelized bandit algorithms in both the classical and quantum settings.

**AutoML Experiment.** In our experiments on *automated machine learning* (AutoML), we use our Q-GP-UCB algorithm to tune the hyperparameters of an SVM for a classification task. Here we consider a Gaussian noise with a variance of $\sigma^2$ since it is more practical and more commonly used in real-world experiments. The results in Fig. 1 (c) show that our Q-GP-UCB again significantly outperforms the classical GP-UCB. Fig. 1 (d) additionally shows the comparisons with LinUCB and Q-LinUCB. The results again corroborate that BO/kernelized bandit algorithms are considerably superior in real-world applications with highly non-linear reward functions in both the classical and quantum settings. These results here further demonstrate the potential of our Q-GP-UCB to lead to quantum speedup in practical applications.

**More Realistic Experiments.** We have additionally tested the performance of our Q-GP-UCB algorithm after accounting for the effect of quantum noise, by incorporating into our `Qiskit` simulations a noise model based on the actual performance of real IBM quantum computers. The results (Fig. 3 in App. J) show that although the addition of quantum noise slightly deteriorates the performance of our Q-GP-UCB, it is still able to significantly outperform classical GP-UCB. Notably, we have additionally performed an experiment *using a real quantum computer* (details in App. J), in which the performance of our Q-GP-UCB, although further worsened compared to the experiment using simulated noise, is still considerably better than classical GP-UCB (Fig. 4, App. J). Also note that the work of [38] has not shown that Q-LinUCB outperforms LinUCB in the presence of simulated quantum noise. Therefore, the consistently superior performances of our Q-GP-UCB over GP-UCB with both simulated quantum noise and a real quantum computer serve as important new support for the potential practical advantages of quantum bandit algorithms.

**Discussion.** In our experimental results (e.g., Fig. 1), our Q-GP-UCB usually has relatively larger regrets in the initial stages but quickly achieves zero regret thereafter (i.e., the curve plateaus), which has also been observed in [38] for the Q-LinUCB algorithm. This is because in the initial stages, our Q-GP-UCB explores a smaller number of unique arms than GP-UCB. However, after the initial exploration, our Q-GP-UCB quickly starts to perform reliable exploitation, because the accurate reward observations achieved thanks to our QMC subroutines allow us to rapidly learn the reward function and hence find the optimal arm.

## 7 Conclusion

We have introduced the first quantum BO algorithm, named Q-GP-UCB. Our Q-GP-UCB achieves a regret upper bound of $\mathcal{O}(\operatorname{poly}\log T)$, which is significantly smaller than the classical regret lower bound of $\Omega(\sqrt{T})$. A limitation of our work is that the regret upper bound of our Q-GP-UCB: $\mathcal{O}((\log T)^{3(d+1)/2})$ (ignoring polylog factors) has a worse dependency on the input dimension $d$ than the regret of the classical GP-UCB: $\mathcal{O}((\log T)^{d+1}\sqrt{T})$. A similar limitation is shared by Q-LinUCB, since its regret has a dependency of $\mathcal{O}(d^{3/2})$ in contrast to the $\mathcal{O}(d)$ of LinUCB. It is an interesting future work to explore potential approaches to remove this extra dependency on $d$. Another limitation of our work is that for the Matérn kernel, we require the reward function to be smooth enough in order to achieve a sub-linear regret upper bound (Sec. 5.7). So, another important future work is to further tighten the regret upper bound of our Q-GP-UCB for the Matérn kernel when the reward function is less smooth, as well as for other kernels encompassing non-smooth functions such as the neural tangent kernel which has been adopted by recent works on neural bandits [13, 42, 44]. Moreover, another interesting future work is to derive regret lower bounds in our setting of quantum kernelized bandits, which can help evaluate the tightness of our regret upper bounds.

## Acknowledgements and Disclosure of Funding

This research/project is supported by the National Research Foundation, Singapore under its AI Singapore Programme (AISG Award No: AISG-PhD/2023-01-039J). DesCartes: this research is supported by the National Research Foundation, Prime Minister's Office, Singapore under its Campus for Research Excellence and Technological Enterprise (CREATE) programme.

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

## A  Analysis of the Weighted GP Posterior (3)

Here we show an alternative view of the weighted GP posterior distribution (3) in the RKHS feature space. The notations in this section will be extensively used in the theoretical analyses in the subsequent sections.

We denote the kernel $k$ by $k(x, x') = \phi(x)^\top \phi(x')$, in which $\phi(x)$ is the potentially infinite-dimensional feature mapping of $x$ in the RKHS of $k$. Define $\Phi_s \triangleq [\phi(x_1), \ldots, \phi(x_s)]^\top$ which is an $s \times \infty$ matrix. Recall that we have defined $W_s \triangleq \text{diag}(\frac{1}{\epsilon_1^2}, \ldots, \frac{1}{\epsilon_s^2})$ which is an $s \times s$ diagonal matrix. Define $V_s \triangleq \lambda I + \sum_{\tau=1}^s \frac{1}{\epsilon_\tau^2}\phi(x_\tau)\phi(x_\tau)^\top = \lambda I + \Phi_s^\top W_s \Phi_s$ which is an $\infty \times \infty$ matrix, and let $V_0 \triangleq \lambda I$. With these notations, it can be easily verified that $K_s = \Phi_s \Phi_s^\top = [k(x_\tau, x_{\tau'})]_{\tau,\tau'=1,\ldots,s}$, and $\widetilde{K}_s = W_s^{1/2}\Phi_s\Phi_s^\top W_s^{1/2} = W_s^{1/2}K_s W_s^{1/2}$. Moreover, we can also easily show that $k_s(x) = \Phi_s \phi(x) = [k(x, x_\tau)]_{\tau=1,\ldots,s}$, $\widetilde{k}_s(x) = W_s^{1/2}\Phi_s \phi(x) = W_s^{1/2}k_s(x)$, and $\widetilde{Y}_s = W_s^{1/2}Y_s$.

We start by deriving an alternative expression for the weighted GP posterior mean $\widetilde{\mu}_s(x)$. To begin with, we have that

$$
\begin{aligned}
\Phi_s^\top W_s^{1/2}(\widetilde{K}_s + \lambda I) &= \Phi_s^\top W_s^{1/2}(W_s^{1/2}\Phi_s\Phi_s^\top W_s^{1/2} + \lambda I) \\
&= \Phi_s^\top W_s^{1/2}W_s^{1/2}\Phi_s\Phi_s^\top W_s^{1/2} + \lambda \Phi_s^\top W_s^{1/2} \\
&= (\Phi_s^\top W_s \Phi_s + \lambda I)\Phi_s^\top W_s^{1/2} \\
&= V_s \Phi_s^\top W_s^{1/2}.
\end{aligned}
$$

Now we multiply both sides by $V_s^{-1}$ on the left and by $(\widetilde{K}_s + \lambda I)^{-1}$ on the right, which leads to

$$
V_s^{-1}\Phi_s^\top W_s^{1/2} = \Phi_s^\top W_s^{1/2}(\widetilde{K}_s + \lambda I)^{-1}
$$

Now we can show that

$$
\begin{aligned}
\widetilde{\mu}_s(x) = \phi(x)^\top V_s^{-1}\Phi_s^\top W_s Y_s &= \phi(x)^\top \Phi_s^\top W_s^{1/2}(\widetilde{K}_s + \lambda I)^{-1}W_s^{1/2}Y_s \\
&= \widetilde{k}_s^\top(x)(\widetilde{K}_s + \lambda I)^{-1}\widetilde{Y}_s.
\end{aligned}
$$

Next, we derive an alternative expression for the weighted GP posterior variance $\widetilde{\sigma}_s^2(x)$. To begin with, we have that

$$
\begin{aligned}
\left(W_s^{-1} + \frac{1}{\lambda}\Phi_s\Phi_s^\top\right)^{-1} &= \left(W_s^{-1/2}\left(I + \frac{1}{\lambda}W_s^{1/2}\Phi_s\Phi_s^\top W_s^{1/2}\right)W_s^{-1/2}\right)^{-1} \\
&= W_s^{1/2}\left(I + \frac{1}{\lambda}W_s^{1/2}\Phi_s\Phi_s^\top W_s^{1/2}\right)^{-1}W_s^{1/2} \\
&= \lambda W_s^{1/2}\left(\lambda I + \widetilde{K}_s\right)^{-1}W_s^{1/2}.
\end{aligned}
\tag{5}
$$

This allows us to show that

$$
\begin{aligned}
\widetilde{\sigma}_s^2(x) &= \lambda \phi(x)^\top V_s^{-1}\phi(x) \\
&= \lambda \phi(x)^\top (\lambda I + \Phi_s^\top W_s \Phi_s)^{-1}\phi(x) \\
&\overset{(a)}{=} \lambda \phi(x)^\top \left(\frac{1}{\lambda}I - \frac{1}{\lambda}\Phi_s^\top(W_s^{-1} + \Phi_s\frac{1}{\lambda}\Phi_s^\top)^{-1}\Phi_s\frac{1}{\lambda}\right)\phi(x) \\
&= \phi(x)^\top \phi(x) - \phi(x)^\top \Phi_s^\top(W_s^{-1} + \frac{1}{\lambda}\Phi_s\Phi_s^\top)^{-1}\Phi_s\frac{1}{\lambda}\phi(x) \\
&\overset{(b)}{=} \phi(x)^\top \phi(x) - \phi(x)^\top \Phi_s^\top W_s^{1/2}\left(\lambda I + \widetilde{K}_s\right)^{-1}W_s^{1/2}\Phi_s\phi(x) \\
&= k(x, x) - \widetilde{k}_s^\top(x)(\widetilde{K}_s + \lambda I)^{-1}\widetilde{k}_s(x),
\end{aligned}
$$

in which we have used the matrix inversion lemma in step $(a)$, and used equation (5) in step $(b)$.

To summarize, we have derived the following alternative expressions for the weighted GP posterior distribution (3):

$$
\begin{aligned}
\widetilde{\mu}_s(x) &= \phi(x)^\top V_s^{-1}\Phi_s^\top W_s Y_s = \widetilde{k}_s^\top(x)(\widetilde{K}_s + \lambda I)^{-1}\widetilde{Y}_s, \\
\widetilde{\sigma}_s^2(x) &= \lambda \phi(x)^\top V_s^{-1}\phi(x) = k(x, x) - \widetilde{k}_s^\top(x)(\widetilde{K}_s + \lambda I)^{-1}\widetilde{k}_s(x).
\end{aligned}
\tag{6}
$$

## B  Proof of Lemma 2

To begin with, we can lower-bound the total number of iterations (i.e., the total number of queries to the quantum oracles) in $m$ stages by

$$\sum_{s=1}^{m} \frac{C_1}{\epsilon_s} \log(\frac{2\overline{m}}{\delta}) \geq \sum_{s=1}^{m} \frac{1}{\epsilon_s} \geq \sqrt{\sum_{s=1}^{m} \frac{1}{\epsilon_s^2}}. \tag{7}$$

The first inequality follows since $C_1 > 1$ (Lemma 1) and $\log(\frac{2\overline{m}}{\delta}) \geq \log(\frac{2}{\delta}) \geq 1$ because we have chosen $\delta \in (0, 2/e]$. Suppose that $\sum_{s=1}^{m} \frac{1}{\epsilon_s^2} > T^2$, then we have that $\sum_{s=1}^{m} \frac{C_1}{\epsilon_s} \log(\frac{2\overline{m}}{\delta}) > T$ which is a contradiction. Therefore, we have that

$$\sum_{s=1}^{m} \frac{1}{\epsilon_s^2} \leq T^2. \tag{8}$$

## C  Proof of Theorem 1

Here we prove the growth rate of $\widetilde{\gamma}_m$, i.e., the weighted information gain. Recall that $W_m = \text{diag}\left(\frac{1}{\epsilon_1^2}, \frac{1}{\epsilon_2^2}, \ldots, \frac{1}{\epsilon_m^2}\right)$, and $\widetilde{\gamma}_m = \frac{1}{2} \log \det\left(I + \lambda^{-1}\widetilde{K}_m\right)$. Our proof here follows closely the proof of Theorem 3 from the work of [35]. We will omit the subscript $m$ as long as the context is clear.

Denote by $K_m = \Phi_m \Phi_m^{\top} = [k(x_i, x_j)]_{i,j=1,\ldots,m}$ the $m \times m$ covariance matrix, and define $\widetilde{K}_m = W_m^{1/2} \Phi_m \Phi_m^{\top} W_m^{1/2} = W_m^{1/2} K_m W_m^{1/2}$. Following Theorem 1 from [35], our kernel $k$ can be denoted as $k(x, x') = \sum_{\tau=1}^{\infty} \lambda_\tau \psi_\tau(x) \psi_\tau(x')$, in which $\{\lambda_\tau\}_{\tau=1,\ldots,\infty}$ and $\{\psi_\tau\}_{\tau=1,\ldots,\infty}$ represent, respectively, the eigenvalues and eigenfunctions of the kernel $k$. We will make use of the projection onto a $D$-dimensional RKHS ($D < \infty$), which consists of the first $D$ features corresponding to the $D$ largest eigenvalues of the kernel $k$. Specifically, define $k_P$ as the $D$-dimensional projection in the RKHS of $k$, and $k_O$ corresponds to the orthogonal element such that $k(\cdot, \cdot) = k_P(\cdot, \cdot) + k_O(\cdot, \cdot)$. That is, following Section 3.2 from the work of [35], define

$$k_P(x, x') = \sum_{\tau=1}^{D} \lambda_\tau \psi_\tau(x) \psi_\tau(x'). \tag{9}$$

We define $K_P \triangleq [k_P(x_i, x_j)]_{i,j=1,\ldots,m}$ and $K_O \triangleq [k_O(x_i, x_j)]_{i,j=1,\ldots,m}$. Similarly, define $\widetilde{K}_P = W_m^{1/2} K_P W_m^{1/2}$ and $\widetilde{K}_O = W_m^{1/2} K_O W_m^{1/2}$. This implies that $K_m = K_P + K_O$ and that $\widetilde{K}_m = \widetilde{K}_P + \widetilde{K}_O$.

Here to be consistent with [35], we use $I_m$ to denote the $m \times m$-dimensional identity matrix. Denote by $\widetilde{I}(\mathbf{y}_m; f)$ the information gain about the function $f$ from the observations $\mathbf{y}_m = [y_1, \ldots, y_m]$. We have that

$$
\begin{aligned}
\widetilde{I}(\mathbf{y}_m; f) &= \frac{1}{2} \log \det\left(I_m + \frac{1}{\lambda}\widetilde{K}_m\right) \\
&= \frac{1}{2} \log \det\left(I_m + \frac{1}{\lambda}\left(\widetilde{K}_P + \widetilde{K}_O\right)\right) \\
&= \frac{1}{2} \log \det\left(\left(I_m + \frac{1}{\lambda}\widetilde{K}_P\right)\left(I_m + \frac{1}{\lambda}\left(I_m + \frac{1}{\lambda}\widetilde{K}_P\right)^{-1}\widetilde{K}_O\right)\right) \\
&= \frac{1}{2} \log \det\left(I_m + \frac{1}{\lambda}\widetilde{K}_P\right) + \frac{1}{2} \log \det\left(I_m + \frac{1}{\lambda}\left(I_m + \frac{1}{\lambda}\widetilde{K}_P\right)^{-1}\widetilde{K}_O\right).
\end{aligned} \tag{10}
$$

The last line follows because $\det(AB) = \det(A)\det(B)$. In the following, we will separately upper-bound the two terms in (10).

We start by upper-bounding the first term in equation (10). Denote by $\boldsymbol{\psi}_D(x)$ the $D$-dimensional feature vector corresponding to the first $D$ principle features of the kernel $k$ (i.e., the features corresponding to the largest $D$ eigenvalues). Define $\Psi_{m,D} \triangleq [\boldsymbol{\psi}_D(x_1), \ldots, \boldsymbol{\psi}_D(x_m)]^\top$ which is an $m \times D$ matrix. Denote by $\Lambda_D = \text{diag}(\lambda_1, \ldots, \lambda_D)$ a $D \times D$-diagonal matrix whose diagonal entries consist of the largest $D$ eigenvalues of the kernel $k$ in descending order. With this definition, we have that $K_P = \Psi_{m,D}\Lambda_D\Psi_{m,D}^\top$ and

$$\widetilde{K}_P = W_m^{1/2}\Psi_{m,D}\Lambda_D\Psi_{m,D}^\top W_m^{1/2}. \tag{11}$$

Also define the gram matrix:

$$\widetilde{G}_m = \Lambda_D^{1/2}\Psi_{m,D}^\top W_m \Psi_{m,D}\Lambda_D^{1/2}. \tag{12}$$

Based on these definitions, we have that

$$\det\left(I_D + \frac{1}{\lambda}\widetilde{G}_m\right) = \det\left(I_m + \frac{1}{\lambda}\widetilde{K}_P\right), \tag{13}$$

which follows from the Weinstein–Aronszajn identity: $\det\left(I_D + AA^\top\right) = \det\left(I_m + A^\top A\right)$ for a $D \times m$ matrix $A$, and we have plugged in $A = \frac{1}{\sqrt{\lambda}}\Lambda_D^{1/2}\Psi_{m,D}^\top W_m^{1/2}$. In addition, the following inequality will also be useful in the subsequent proof:

$$\left\|\Lambda_D^{1/2}\boldsymbol{\psi}_D(x)\right\|_2^2 = \sum_{\tau=1}^{D}\lambda_\tau\psi_\tau^2(x) = k_P(x,x) \leq k(x,x) \leq 1, \tag{14}$$

in which we have made use of the definition of $k_P$ from (9), and our assumption (w.l.o.g.) that $k(x,x') \leq 1, \forall x, x'$ (Sec. 3.1). In the following, we will denote the trace of a matrix $A$ by $\text{Tr}(A)$. Next, we have that

$$\begin{aligned}
\text{Tr}\left(I_D + \frac{1}{\lambda}\widetilde{G}_m\right) &= D + \frac{1}{\lambda}\text{Tr}\left(\widetilde{G}_m\right) \\
&= D + \frac{1}{\lambda}\text{Tr}\left(\Lambda_D^{1/2}\Psi_{m,D}^\top W_m \Psi_{m,D}\Lambda_D^{1/2}\right) \\
&\overset{(a)}{=} D + \frac{1}{\lambda}\text{Tr}\left(\Lambda_D^{1/2}\left[\sum_{s=1}^{m}\frac{1}{\epsilon_s^2}\boldsymbol{\psi}_D(x_s)\boldsymbol{\psi}_D^\top(x_s)\right]\Lambda_D^{1/2}\right) \\
&= D + \frac{1}{\lambda}\text{Tr}\left(\sum_{s=1}^{m}\frac{1}{\epsilon_s^2}\Lambda_D^{1/2}\boldsymbol{\psi}_D(x_s)\boldsymbol{\psi}_D^\top(x_s)\Lambda_D^{1/2}\right) \\
&= D + \frac{1}{\lambda}\sum_{s=1}^{m}\frac{1}{\epsilon_s^2}\text{Tr}\left(\Lambda_D^{1/2}\boldsymbol{\psi}_D(x_s)\boldsymbol{\psi}_D^\top(x_s)\Lambda_D^{1/2}\right) \\
&\overset{(b)}{=} D + \frac{1}{\lambda}\sum_{s=1}^{m}\frac{1}{\epsilon_s^2}\text{Tr}\left(\boldsymbol{\psi}_D^\top(x_s)\Lambda_D^{1/2}\Lambda_D^{1/2}\boldsymbol{\psi}_D(x_s)\right) \\
&= D + \frac{1}{\lambda}\sum_{s=1}^{m}\frac{1}{\epsilon_s^2}\boldsymbol{\psi}_D^\top(x_s)\Lambda_D^{1/2}\Lambda_D^{1/2}\boldsymbol{\psi}_D(x_s) \\
&= D + \frac{1}{\lambda}\sum_{s=1}^{m}\frac{1}{\epsilon_s^2}\left\|\Lambda_D^{1/2}\boldsymbol{\psi}_D(x_s)\right\|_2^2 \\
&\overset{(c)}{\leq} D + \frac{1}{\lambda}\sum_{s=1}^{m}\frac{1}{\epsilon_s^2},
\end{aligned} \tag{15}$$

in which $(a)$ follows because $\Psi_{m,D}^\top W_m \Psi_{m,D} = \sum_{s=1}^{m}\frac{1}{\epsilon_s^2}\boldsymbol{\psi}_D(x_s)\boldsymbol{\psi}_D^\top(x_s)$, $(b)$ has made use of the cyclic property of the trace operator, and $(c)$ follows from (14) above.

Also note that for a positive definite matrix $P \in \mathbb{R}^{n \times n}$, we have that

$$\log \det(P) \le n \log \frac{\text{Tr}(P)}{n}. \tag{16}$$

Therefore, we have that

$$\log \det \left( I_m + \frac{1}{\lambda} \widetilde{K}_P \right) = \log \det \left( I_D + \frac{1}{\lambda} \widetilde{G}_m \right)$$

$$\le D \log \frac{D + \frac{1}{\lambda} \sum_{s=1}^m \frac{1}{\epsilon_s^2}}{D} = D \log \left( 1 + \frac{1}{\lambda D} \sum_{s=1}^m \frac{1}{\epsilon_s^2} \right). \tag{17}$$

The first equality follows from (13), and the inequality results from (16) and (15).

Now we upper-bound the second term in equation (10). To begin with, we have that

$$\text{Tr} \left( \left( I_m + \frac{1}{\lambda} \widetilde{K}_P \right)^{-1} \widetilde{K}_O \right) \le \text{Tr} \left( \widetilde{K}_O \right), \tag{18}$$

which is because the matrix $\left( I_m + \frac{1}{\lambda} \widetilde{K}_P \right)^{-1}$ is positive semi-definite (PSD) whose largest eigenvalue is upper-bounded by 1, and $\text{Tr}(P_1 P_2) \le \lambda_{P_1} \text{Tr}(P_2)$ where $P_1$ and $P_2$ are PSD matrices and $\lambda_{P_1}$ is the largest eigenvalue of $P_1$. Next, define $\delta_D \triangleq \sum_{\tau=D+1}^\infty \lambda_\tau \psi^2$ where $|\phi_\tau(x)| \le \psi, \forall x, \tau$. Then we immediately have that $k_O(x, x') = \sum_{\tau=D+1}^\infty \lambda_\tau \phi_\tau(x) \phi_\tau(x') \le \delta_D, \forall x, x'$. Therefore, we have that

$$\text{Tr} \left( \widetilde{K}_O \right) = \sum_{s=1}^m \frac{1}{\epsilon_s^2} k_O(x_s, x_s) \le \delta_D \sum_{s=1}^m \frac{1}{\epsilon_s^2}. \tag{19}$$

As a result, we have that

$$\text{Tr} \left( I_m + \frac{1}{\lambda} \left( I_m + \frac{1}{\lambda} \widetilde{K}_P \right)^{-1} \widetilde{K}_O \right) \le m + \frac{1}{\lambda} \delta_D \sum_{s=1}^m \frac{1}{\epsilon_s^2}, \tag{20}$$

in which the inequality follows from (18) and (19).

Next, again making use of equation (16) and incorporating the upper bound on the trace from (20), we have that

$$\log \det \left( I_m + \frac{1}{\lambda} \left( I_m + \frac{1}{\lambda} \widetilde{K}_P \right)^{-1} \widetilde{K}_O \right) \le m \log \frac{m + \frac{1}{\lambda} \delta_D \sum_{s=1}^m \frac{1}{\epsilon_s^2}}{m}$$

$$\le m \log \left( 1 + \frac{\delta_D}{m\lambda} \sum_{s=1}^m \frac{1}{\epsilon_s^2} \right)$$

$$\le m \times \frac{\delta_D}{m\lambda} \sum_{s=1}^m \frac{1}{\epsilon_s^2}$$

$$\le \left( \sum_{s=1}^m \frac{1}{\epsilon_s^2} \right) \frac{\delta_D}{\lambda}. \tag{21}$$

in which the second last step is because $\log(1 + z) \le z, \forall z \in \mathbb{R}$.

Finally, combining the upper bounds from 17 and 21, equation (10) can be upper-bounded by

$$\widetilde{I}(\mathbf{y}_m; f) \le \frac{1}{2} D \log \left( 1 + \frac{\sum_{s=1}^m \frac{1}{\epsilon_s^2}}{D\lambda} \right) + \frac{1}{2} \left( \sum_{s=1}^m \frac{1}{\epsilon_s^2} \right) \frac{\delta_D}{\lambda}. \tag{22}$$

Therefore, we have that

$$\widetilde{\gamma}_m = \widetilde{I}(\mathbf{y}_m; f) = \frac{1}{2} \log \det \left( I_m + \frac{1}{\lambda} \widetilde{K}_m \right) \le \frac{1}{2} D \log(1 + \frac{\sum_{s=1}^m \frac{1}{\epsilon_s^2}}{D\lambda}) + \frac{1}{2} \left( \sum_{s=1}^m \frac{1}{\epsilon_s^2} \right) \frac{\delta_D}{\lambda}. \tag{23}$$

Also recall that for the standard unweighted maximum information gain $\gamma_T$, we have that [35]

$$\gamma_T = \frac{1}{2}\log\det\left(I_T + \frac{1}{\lambda}K_T\right) \leq \frac{1}{2}D\log(1 + \frac{T}{D\lambda}) + \frac{1}{2}T\frac{\delta_D}{\lambda}. \tag{24}$$

Therefore, the upper bound on the weighted information gain $\widetilde{\gamma}_m$ is obtained by replacing the $T$ from the upper bound on standard maximum information gain $\gamma_T$ by $\sum_{s=1}^{m}\frac{1}{\epsilon_s^2}$. This is formalized by the following Theorem.

**Theorem 8.** *[More formal statement of Theorem 1 (a)] Given $\mathcal{X}_m \triangleq \{x_1, \ldots, x_m\}$, then we have that*

$$\widetilde{\gamma}_m \leq \frac{1}{2}D\log(1 + \frac{\sum_{s=1}^{m}\frac{1}{\epsilon_s^2}}{D\lambda}) + \frac{1}{2}\left(\sum_{s=1}^{m}\frac{1}{\epsilon_s^2}\right)\frac{\delta_D}{\lambda},$$

$$\gamma_{\sum_{s=1}^{m}1/\epsilon_s^2} \leq \frac{1}{2}D\log(1 + \frac{\sum_{s=1}^{m}\frac{1}{\epsilon_s^2}}{D\lambda}) + \frac{1}{2}\left(\sum_{s=1}^{m}\frac{1}{\epsilon_s^2}\right)\frac{\delta_D}{\lambda}.$$

Lastly, note that it has been shown that $\gamma_T = \mathcal{O}(d\log T)$ for the linear kernel and $\gamma_T = \mathcal{O}((\log T)^{d+1})$ for the SE kernel, Therefore, plugging in Lemma 2, which allows us to upper-bound $\sum_{s=1}^{m}1/\epsilon_s^2$ by $T^2$, leads to Corollary 1.

## D   Proof of Theorem 2

Our proof here follows the outline of the proof of Lemma 2 from the work of [38]. To begin with, we show that $\det(V_{\tau+1}) = 2\det(V_\tau), \forall \tau = 0, \ldots, m-1$, i.e., the amount of collected information is doubled after every stage.

$$\begin{aligned}
\det(V_{\tau+1}) &= \det\left(\lambda I + \sum_{j=1}^{\tau+1}\frac{1}{\epsilon_j^2}\phi(x_j)\phi(x_j)^\top\right) \\
&= \det\left(\lambda I + \sum_{j=1}^{\tau}\frac{1}{\epsilon_j^2}\phi(x_j)\phi(x_j)^\top + \frac{1}{\epsilon_{\tau+1}^2}\phi(x_{\tau+1})\phi(x_{\tau+1})^\top\right) \\
&= \det\left(V_\tau + \frac{1}{\epsilon_{\tau+1}^2}\phi(x_{\tau+1})\phi(x_{\tau+1})^\top\right) \\
&= \det\left(V_\tau^{1/2}\left(I + \frac{1}{\epsilon_{\tau+1}^2}V_\tau^{-1/2}\phi(x_{\tau+1})\phi(x_{\tau+1})^\top V_\tau^{-1/2}\right)V_\tau^{1/2}\right) \\
&= \det(V_\tau)\det\left(I + \frac{1}{\epsilon_{\tau+1}^2}V_\tau^{-1/2}\phi(x_{\tau+1})\phi(x_{\tau+1})^\top V_\tau^{-1/2}\right) \\
&\overset{(a)}{=} \det(V_\tau)\left(1 + \frac{1}{\epsilon_{\tau+1}^2}\phi(x_{\tau+1})^\top V_\tau^{-1/2}V_\tau^{-1/2}\phi(x_{\tau+1})\right) \\
&= \det(V_\tau)\left(1 + \frac{1}{\epsilon_{\tau+1}^2}\left\|\phi(x_{\tau+1})\right\|_{V_\tau^{-1}}^2\right) \\
&\overset{(b)}{=} \det(V_\tau)\left(1 + \frac{\widetilde{\sigma}_\tau^2(x_{\tau+1})/\lambda}{\widetilde{\sigma}_\tau^2(x_{\tau+1})/\lambda}\right) \\
&= 2\det(V_\tau).
\end{aligned} \tag{25}$$

Step $(a)$ has made use of the matrix determinant lemma: $\det(I + ab^\top) = \det(I + a^\top b)$, and $(b)$ follows from the expression of the weighted GP posterior variance (6) and our choice of $\epsilon_{\tau+1} =$

$\widetilde{\sigma}_\tau(x_{\tau+1})/\sqrt{\lambda}$ (line 3 of Algo. 1). An immediate consequence of (25) is that $\det(V_m) = 2^m \det(V_0)$. Recall we have defined earlier that $V_0 = \lambda I$ (App. A).

Recall that following the notations of App. A, we have that $V_m = \lambda I + \Phi_m^\top W_m \Phi_m$. Next, we can also show that

$$
\begin{aligned}
\log \frac{\det(V_m)}{\det(V_0)} &= \log \frac{\det(\lambda I + \Phi_m^\top W_m \Phi_m)}{\det(\lambda I)} \\
&= \log \det \left( I + \frac{1}{\lambda}(\Phi_m^\top W_m^{1/2})(W_m^{1/2}\Phi_m) \right) \\
&\overset{(a)}{=} \log \det \left( I + \frac{1}{\lambda} W_m^{1/2}\Phi_m \Phi_m^\top W_m^{1/2} \right) \\
&= \log \det \left( I + \frac{1}{\lambda}\widetilde{K}_m \right) \\
&= 2\widetilde{\gamma}_m,
\end{aligned}
\tag{26}
$$

in which step $(a)$ has again made use of the Weinstein–Aronszajn identity (similar to (13)), and the last two equalities have simply plugged in the expressions for $\widetilde{K}_s = W_s^{1/2}\Phi_s \Phi_s^\top W_s^{1/2}$ (App. A), and $\widetilde{\gamma}_m = \frac{1}{2}\log\det(I + \lambda^{-1}\widetilde{K}_m)$ (Sec. 5.1).

According to Corollary 1, for the linear kernel, we have that there exists an absolute constant $C > 0$ such that $\widetilde{\gamma}_m \leq Cd\log\left(\sum_{s=1}^m \frac{1}{\epsilon_s^2}\right)$. Combining equations (25) and (26), we can show that

$$
m\log 2 = \log \frac{\det(V_m)}{\det(V_0)} = 2\widetilde{\gamma}_m \leq 2Cd\log\left(\sum_{s=1}^m \frac{1}{\epsilon_s^2}\right).
\tag{27}
$$

This allows us to show that

$$
\sum_{s=1}^m \frac{1}{\epsilon_s^2} \geq (2^m)^{1/(2Cd)}.
\tag{28}
$$

Note that in every stage $s$, we query the quantum oracle for $\frac{C_1}{\epsilon_s}\log(\frac{2\overline{m}}{\delta})$ times, in which $C_1 > 1$, $\delta \in (0, 2/e]$ and $\overline{m}$ is an upper bound on the total number of stages. An immediate consequence is that $\log(\frac{2\overline{m}}{\delta}) \geq \log(\frac{2}{\delta}) \geq 1$. Therefore, the total number of iterations can be analyzed as

$$
\sum_{s=1}^m \frac{C_1}{\epsilon_s}\log(\frac{2\overline{m}}{\delta}) \geq \sum_{s=1}^m \frac{1}{\epsilon_s} \geq \sqrt{\sum_{s=1}^m \frac{1}{\epsilon_s^2}} \geq \sqrt{(2^m)^{1/(2Cd)}}.
\tag{29}
$$

Now we derive an upper bound on $m$ by contradiction. Suppose $m > \frac{2Cd}{\log 2}\log T$, this immediately implies that

$$
\sqrt{(2^m)^{1/(2Cd)}} > T.
\tag{30}
$$

This equation, combined with (29), implies that $\sum_{s=1}^m \frac{C_1}{\epsilon_s}\log(\frac{2\overline{m}}{\delta}) > T$, which is a contradiction. Therefore, we have that

$$
m \leq \frac{2}{\log 2}Cd\log T = \mathcal{O}(d\log T).
\tag{31}
$$

That is, for the linear kernel, our algorithm runs for at most $\mathcal{O}(d\log T)$ stages, which matches the upper bound on the total number of stages for the quantum linear UCB (Q-LinUCB) algorithm, as proved by Lemma 2 from the work of [38].

Next, for the SE kernel, Corollary 1 shows that there exists an absolute constant $C > 0$ such that $\widetilde{\gamma}_m \leq C\log^{d+1}\left(\sum_{s=1}^m \frac{1}{\epsilon_s^2}\right)$. Again combining equations (25) and (26), we can show that

$$
m\log 2 = \log \frac{\det(V_m)}{\det(V_0)} = 2\widetilde{\gamma}_m \leq 2C\log^{d+1}\left(\sum_{s=1}^m \frac{1}{\epsilon_s^2}\right).
\tag{32}
$$

This allows us to show that

$$\sum_{s=1}^{m} \frac{1}{\epsilon_s^2} \geq \exp\left(\left(\frac{m \log 2}{2C}\right)^{1/(d+1)}\right). \tag{33}$$

Now again we derive an upper bound on $m$ by contradiction. Suppose $m > \frac{2C}{\log 2}(2 \log T)^{d+1}$, then this implies that

$$\sqrt{\exp\left(\left(\frac{m \log 2}{2C}\right)^{1/(d+1)}\right)} > T. \tag{34}$$

This, combined with (29), implies that $\sum_{s=1}^{m} \frac{C_1}{\epsilon_s} \log(\frac{2\overline{m}}{\delta}) > T$ which is a contradiction. Therefore

$$m \leq \frac{2C}{\log 2}(2 \log T)^{d+1} = \mathcal{O}\left((\log T)^{d+1}\right) \tag{35}$$

for the SE kernel. This completes the proof.

## E    Proof of Theorem 3

Our proof here follows closely the proof of Theorem 2 from [9]. Denote by $\zeta_s$ the observation noise for the observation in stage $s$: $y_s = f(x_s) + \zeta_s$. Define $\zeta_{1:s} \triangleq [\zeta_1, \ldots, \zeta_s]^\top$ which is an $s \times 1$ vector. Denote by $\mathcal{F}_s$ the $\sigma$-algebra: $\mathcal{F}_s = \{x_1, \ldots, x_{s+1}, \zeta_1, \ldots, \zeta_s\}$. With this definition, we have that $x_s$ is $\mathcal{F}_{s-1}$-measurable, and $\zeta_s$ is $\mathcal{F}_s$-measurable. Note that we have used $\phi(\cdot)$ to denote the RKHS feature map of the kernel $k$: $k(x, x') = \phi(x)^\top \phi(x')$, and here we let $\phi(x) = k(x, \cdot)$. Then using the reproducing property, we have that $f(x) = \langle \phi(x), f \rangle_k = \phi(x)^\top f$, in which the inner product denotes the inner product induced by the RKHS of $k$.

To begin with, we have that

$$\Phi_s^\top W_s^{1/2}(W_s^{1/2}\Phi_s\Phi_s^\top W_s^{1/2} + \lambda I)^{-1} = (\Phi_s^\top W_s\Phi_s + \lambda I)^{-1}\Phi_s^\top W_s^{1/2}, \tag{36}$$

which follows from noting that $(\Phi_s^\top W_s\Phi_s + \lambda I)\Phi_s^\top W_s^{1/2} = \Phi_s^\top W_s^{1/2}(W_s^{1/2}\Phi_s\Phi_s^\top W_s^{1/2} + \lambda I)$, and then multiplying by $(\Phi_s^\top W_s\Phi_s + \lambda I)^{-1}$ on the left and by $(W_s^{1/2}\Phi_s\Phi_s^\top W_s^{1/2} + \lambda I)^{-1}$ on the right.

Now let $f_s \triangleq [f(x_k)]_{k=1,\ldots,s}^\top = \Phi_s f$ which is an $s \times 1$-dimensional vector, and $\widetilde{f}_s \triangleq W_s^{1/2} f_s = W_s^{1/2}\Phi_s f$. Next, we can prove that

$$
\begin{aligned}
|f(x) - \widetilde{k}_s^\top(x)(\widetilde{K}_s + \lambda I)^{-1}\widetilde{f}_s| &= |f(x) - \phi(x)^\top \Phi_s^\top W_s^{1/2}\left(W_s^{1/2}\Phi_s\Phi_s^\top W_s^{1/2} + \lambda I\right)^{-1}\widetilde{f}_s| \\
&\overset{(a)}{=} |f(x) - \phi(x)^\top(\Phi_s^\top W_s\Phi_s + \lambda I)^{-1}\Phi_s^\top W_s^{1/2}W_s^{1/2}\Phi_s f| \\
&= |\phi(x)^\top f - \phi(x)^\top(\Phi_s^\top W_s\Phi_s + \lambda I)^{-1}\Phi_s^\top W_s\Phi_s f| \\
&= |\lambda\phi(x)^\top(\Phi_s^\top W_s\Phi_s + \lambda I)^{-1}f| \\
&\leq \|f\|_k \left\|\lambda\phi(x)^\top(\Phi_s^\top W_s\Phi_s + \lambda I)^{-1}\right\|_k \\
&= \|f\|_k \sqrt{\lambda\phi(x)^\top(\Phi_s^\top W_s\Phi_s + \lambda I)^{-1}\lambda I(\Phi_s^\top W_s\Phi_s + \lambda I)^{-1}\phi(x)} \\
&\leq \|f\|_k \sqrt{\lambda\phi(x)^\top(\Phi_s^\top W_s\Phi_s + \lambda I)^{-1}(\lambda I + \Phi_s^\top W_s\Phi_s)(\Phi_s^\top W_s\Phi_s + \lambda I)^{-1}\phi(x)} \\
&\leq \|f\|_k \sqrt{\lambda\phi(x)^\top(\Phi_s^\top W_s\Phi_s + \lambda I)^{-1}\phi(x)} \\
&\overset{(b)}{\leq} B\widetilde{\sigma}_s(x),
\end{aligned} \tag{37}
$$

in which step $(a)$ has made use of (36), and step $(b)$ follows from our assumption that $\|f\|_k \leq B$ (Sec. 3.1) and the expression for the weighted GP posterior variance (6). Next, we have that

$$
\begin{aligned}
|\widetilde{k}_s^\top(x)(\widetilde{K}_s + \lambda I)^{-1}W_s^{1/2}\zeta_{1:s}| &= |\phi(x)^\top \Phi_s^\top W_s^{1/2}\left(W_s^{1/2}\Phi_s\Phi_s^\top W_s^{1/2} + \lambda I\right)^{-1} W_s^{1/2}\zeta_{1:s}| \\
&\overset{(a)}{=} |\phi(x)^\top(\Phi_s^\top W_s\Phi_s + \lambda I)^{-1}\Phi_s^\top W_s^{1/2}W_s^{1/2}\zeta_{1:s}| \\
&= |\phi(x)^\top(\Phi_s^\top W_s\Phi_s + \lambda I)^{-1}\Phi_s^\top W_s\zeta_{1:s}| \\
&\leq \left\|(\Phi_s^\top W_s\Phi_s + \lambda I)^{-1/2}\phi(x)\right\|_k \left\|(\Phi_s^\top W_s\Phi_s + \lambda I)^{-1/2}\Phi_s^\top W_s\zeta_{1:s}\right\|_k \\
&= \sqrt{\phi(x)^\top(\Phi_s^\top W_s\Phi_s + \lambda I)^{-1}\phi(x)}\sqrt{(\Phi_s^\top W_s\zeta_{1:s})^\top(\Phi_s^\top W_s\Phi_s + \lambda I)^{-1}\Phi_s^\top W_s\zeta_{1:s}} \\
&= \frac{1}{\sqrt{\lambda}}\widetilde{\sigma}_s(x)\sqrt{\zeta_{1:s}^\top W_s\Phi_s(\Phi_s^\top W_s\Phi_s + \lambda I)^{-1}\Phi_s^\top W_s^{1/2}W_s^{1/2}\zeta_{1:s}} \\
&\overset{(b)}{=} \frac{1}{\sqrt{\lambda}}\widetilde{\sigma}_s(x)\sqrt{\zeta_{1:s}^\top W_s\Phi_s\Phi_s^\top W_s^{1/2}(W_s^{1/2}\Phi_s\Phi_s^\top W_s^{1/2} + \lambda I)^{-1}W_s^{1/2}\zeta_{1:s}} \\
&= \frac{1}{\sqrt{\lambda}}\widetilde{\sigma}_s(x)\sqrt{\zeta_{1:s}^\top W_s^{1/2}W_s^{1/2}\Phi_s\Phi_s^\top W_s^{1/2}(W_s^{1/2}\Phi_s\Phi_s^\top W_s^{1/2} + \lambda I)^{-1}W_s^{1/2}\zeta_{1:s}} \\
&= \frac{1}{\sqrt{\lambda}}\widetilde{\sigma}_s(x)\sqrt{\zeta_{1:s}^\top W_s^{1/2}\widetilde{K}_s(\widetilde{K}_s + \lambda I)^{-1}W_s^{1/2}\zeta_{1:s}},
\end{aligned}
\tag{38}
$$

in which $(a)$ and $(b)$ follow from (36). Since $\widetilde{\mu}_s(x) = \widetilde{k}_s^\top(x)(\widetilde{K}_s + \lambda I)^{-1}\widetilde{Y}_s$ and $\widetilde{Y}_s = \widetilde{f}_s + W_s^{1/2}\zeta_{1:s}$, then the two equations above combine to tell us that

$$
\begin{aligned}
|f(x) - \widetilde{\mu}_s(x)| &\leq \widetilde{\sigma}_s(x)\left(B + \frac{1}{\sqrt{\lambda}}\sqrt{\zeta_{1:s}^\top W_s^{1/2}\widetilde{K}_s(\widetilde{K}_s + \lambda I)^{-1}W_s^{1/2}\zeta_{1:s}}\right) \\
&\leq \widetilde{\sigma}_s(x)\left(B + \sqrt{\zeta_{1:s}^\top W_s^{1/2}\widetilde{K}_s(\widetilde{K}_s + \lambda I)^{-1}W_s^{1/2}\zeta_{1:s}}\right),
\end{aligned}
\tag{39}
$$

in which the second inequality follows since $\lambda = 1 + \eta > 1$. Again following [9], we apply a few more steps of transformations. Note that for an invertible $K$, we have that $K(K + I)^{-1} = \left((K + I)K^{-1}\right)^{-1} = \left(I + K^{-1}\right)^{-1}$. Substituting $K = \widetilde{K}_s + \eta I$, we have that

$$
(\widetilde{K}_s + \eta I)(\widetilde{K}_s + (\eta + 1)I)^{-1} = \left(I + (\widetilde{K}_s + \eta I)^{-1}\right)^{-1}.
$$

Next, we have that

$$
\begin{aligned}
\zeta_{1:s}^\top W_s^{1/2}\widetilde{K}_s(\widetilde{K}_s + \lambda I)^{-1}W_s^{1/2}\zeta_{1:s} &\leq \zeta_{1:s}^\top W_s^{1/2}(\widetilde{K}_s + \eta I)(\widetilde{K}_s + (1 + \eta)I)^{-1}W_s^{1/2}\zeta_{1:s} \\
&= \zeta_{1:s}^\top W_s^{1/2}\left((\widetilde{K}_s + \eta I)^{-1} + I\right)^{-1}W_s^{1/2}\zeta_{1:s} \\
&= \left\|W_s^{1/2}\zeta_{1:s}\right\|_{((\widetilde{K}_s + \eta I)^{-1} + I)^{-1}}^2.
\end{aligned}
$$

This allows us to further re-write (39) as

$$
|f(x) - \widetilde{\mu}_s(x)| \leq \widetilde{\sigma}_s(x)\left(B + \left\|W_s^{1/2}\zeta_{1:s}\right\|_{((\widetilde{K}_s + \eta I)^{-1} + I)^{-1}}\right).
\tag{40}
$$

Recall that $W_s = \text{diag}\left(\frac{1}{\epsilon_1^2}, \frac{1}{\epsilon_2^2}, \ldots, \frac{1}{\epsilon_s^2}\right)$, therefore, $W_s^{1/2}\zeta_{1:s} = [\zeta_k \frac{1}{\epsilon_k}]_{k=1,\ldots,s}^\top$. Of note, since we have that $|f(x_s) - y_s| \leq \epsilon_s, \forall s = 1, \ldots, m$ (with high probability, refer to the beginning of Sec. 5), so, the noise $\zeta_s = y_s - f(x_s)$ is bounded: $|\zeta_s| \leq \epsilon_s, \forall s = 1, \ldots, m$. In other words, $\zeta_s$ is $\epsilon_s$-sub-Gaussian.

To begin with, note that $\epsilon_k$ is $\mathcal{F}_{k-1}$-measurable. This can be seen recursively: conditioned on $\{x_1\}$, $\epsilon_1 = \widetilde{\sigma}_0(x_1)/\sqrt{\lambda}$ is a deterministic constant (i.e., predictable); conditioned on $\{x_1, x_2\}$, $\widetilde{\sigma}_1(\cdot)$ is predictable because it depends on $x_1$ and $\epsilon_1$ (via the weight $\frac{1}{\epsilon_1^2}$), and hence $\epsilon_2 = \widetilde{\sigma}_1(x_2)/\sqrt{\lambda}$ is predictable conditioned on $\{x_1, x_2\}$. By induction, conditioned on $\{x_1, \ldots, x_k\}$, $\epsilon_k = \widetilde{\sigma}_{k-1}(x_k)/(\sqrt{\lambda})$ is predictable. Therefore, $\epsilon_k$ is $\mathcal{F}_{k-1}$-measurable, because $\mathcal{F}_{k-1} = \{x_1, \ldots, x_k, \zeta_1, \ldots, \zeta_{k-1}\}$.

Next, note that since $\zeta_k$ is $\mathcal{F}_k$-measurable, we have that $\frac{\zeta_k}{\epsilon_k}$ is $\mathcal{F}_k$-measurable. Moreover, conditioned on $\mathcal{F}_{k-1}$ (i.e., $\epsilon_k$ is a predictable), $\frac{\zeta_k}{\epsilon_k}$ is 1-sub-Gaussian, because $\zeta_k$ is $\epsilon_k$-sub-Gaussian as discussed above. To summarize, we have that *every element $\frac{\zeta_k}{\epsilon_k}$ of this noise vector $W_s^{1/2}\zeta_{1:s}$ is (a) $\mathcal{F}_k$-measurable and (b) 1-sub-Gaussian conditionally on $\mathcal{F}_{k-1}$.* Therefore, we can apply Theorem 1 of [9] to 1-sub-Gaussian noise, to show that

$$\left\| W_s^{1/2}\zeta_{1:s} \right\|_{((\widetilde{K}_s+\eta I)^{-1}+I)^{-1}} \leq \sqrt{2\log\frac{\sqrt{\det((1+\eta)I+\widetilde{K}_s)}}{\delta}} \tag{41}$$

with probability of at least $1 - \delta$. When applying Theorem 1 of [9], we have also replaced the original unweighted covariance matrix $K_s$ by the corresponding weighted covariance matrix $\widetilde{K}_s = W_s^{1/2}K_sW_s^{1/2}$. This is possible because every $\epsilon_k$ is $\mathcal{F}_{k-1}$-measurable. More concretely, in the proof of Theorem 1 of [9], when defining the super-martingale $\{M_t\}_t$ which is conditioned on $\mathcal{F}_\infty$, we only need to replace the covariance matrix $K_t$ (for the multivariate Gaussian distribution of $h(x_1), \ldots, h(x_t)$) by $\widetilde{K}_t = W_t^{1/2}K_tW_t^{1/2}$, because both the original $K_t$ and our $\widetilde{K}_t$ only require conditioning on $\{x_1, \ldots, x_t\}$.

Combining the two equations above allows us to show that

$$|f(x) - \widetilde{\mu}_s(x)| \leq \widetilde{\sigma}_s(x)\left(B + \sqrt{2\log\frac{\sqrt{\det((1+\eta)I+\widetilde{K}_s)}}{\delta}}\right) \tag{42}$$

with probability of $\geq 1 - \delta$. Next, we can further upper-bound the log determinant term:

$$\begin{aligned}
\log\det\left((1+\eta)I+\widetilde{K}_s\right) &\overset{(a)}{\leq} s\log(1+\eta) + \log\det\left(I + \frac{1}{1+\eta}\widetilde{K}_s\right) \\
&\overset{(b)}{\leq} \log\det\left(I + \frac{1}{\lambda}\widetilde{K}_s\right) + s\eta \\
&\overset{(c)}{=} 2\widetilde{\gamma}_s + s\eta \\
&\overset{(d)}{\leq} 2\widetilde{\gamma}_s + 2,
\end{aligned} \tag{43}$$

in which $(a)$ follows since $\det(AB) = \det(A)\det(B)$ and $\det(cA) \leq c^s\det(A)$ for a scalar $c$ and an $s \times s$-matrix $A$, $(b)$ has made use of $\log(1+z) \leq z$, $(c)$ has made use of the definition of $\gamma_s$, and $(d)$ follows because $\eta = 2/T$ and $s \leq T$. This eventually allows us to show that

$$\begin{aligned}
|f(x) - \widetilde{\mu}_s(x)| &\leq \widetilde{\sigma}_s(x)\left(B + \sqrt{2\log\frac{\sqrt{\det((1+\eta)I+\widetilde{K}_s)}}{\delta}}\right) \\
&\leq \widetilde{\sigma}_s(x)\left(B + \sqrt{2(\widetilde{\gamma}_s + 1 + \log(1/\delta))}\right),
\end{aligned} \tag{44}$$

Of note, although the work of [15] has also adopted weighted GP regression in a similar way to our (3), our proof technique here cannot be applied in their analysis. This is because the work of [15] has chosen the weight to be $w_s = \eta^{-s}$ for $\eta \in (0, 1)$ and hence $W_s = \text{diag}[w_1, \ldots, w_s]$. Therefore, every element of the scaled noise vector $W_s^{1/2}\zeta_{1:s}$ (i.e., $\zeta_k\sqrt{w_k}$) is not guaranteed to be sub-Gaussian. As a result, this makes it infeasible for them to apply the self-normalizing concentration inequality from Theorem 1 of [9], i.e., our (41) above does not hold in their case. The work of [15] has come up with other novel proof techniques suited to their setting. Therefore, our proof here only works in our problem setting of quantum BO.

# F   Proof of Theorem 4

To begin with, we can show that

$$r_s = f(x^*) - f(x_s) \overset{(a)}{\leq} \widetilde{\mu}_{s-1}(x^*) + \beta_s \widetilde{\sigma}_{s-1}(x^*) - f(x_s)$$

$$\overset{(b)}{\leq} \widetilde{\mu}_{s-1}(x_s) + \beta_s \widetilde{\sigma}_{s-1}(x_s) - f(x_s) \tag{45}$$

$$\overset{(c)}{\leq} 2\beta_s \widetilde{\sigma}_{s-1}(x_s) \overset{(d)}{=} 2\beta_s \epsilon_s \sqrt{\lambda},$$

in which $(a)$ and $(c)$ follow from Theorem 3, $(b)$ results from our policy for selecting $x_s$ (line 2 of Algo. 1), and $(d)$ follows from our design of $\epsilon_s = \widetilde{\sigma}_{s-1}(x_s)/\sqrt{\lambda}$ (line 3 of Algo. 1). As a result, the total regret in stage $s$ can be upper bounded by:

$$\frac{C_1}{\epsilon_s} \log \frac{2\overline{m}}{\delta} \times 2\beta_s \epsilon_s \sqrt{\lambda} = 2C_1 \beta_s \sqrt{\lambda} \log \frac{2\overline{m}}{\delta}. \tag{46}$$

Then an upper bound on the total cumulative regret can be obtained by summing up the regrets from all $m$ stages:

$$R_T \leq \sum_{s=1}^{m} 2C_1 \beta_s \sqrt{\lambda} \log \frac{2\overline{m}}{\delta} \leq m 2C_1 \beta_m \sqrt{\lambda} \log \frac{2\overline{m}}{\delta} = \mathcal{O}\left( m \beta_m \log \frac{m}{\delta} \right). \tag{47}$$

For the linear kernel, recall from Theorem 3 and Corollary 1 that $\beta_m = \mathcal{O}(\sqrt{\widetilde{\gamma}_{m-1} + \log(1/\delta)}) = \mathcal{O}(\sqrt{d \log T + \log(1/\delta)})$. Also recall that Theorem 2 tells us that $m \leq \overline{m} = \mathcal{O}(d \log T)$. Therefore, for the linear kernel,

$$R_T = \mathcal{O}\left( d \log T \sqrt{d \log T + \log(1/\delta)} \log \frac{d \log T}{\delta} \right) = \mathcal{O}\left( d^{3/2} \log^{3/2} T \log(d \log T) \right), \tag{48}$$

in which we have ignored the dependency on $\log(1/\delta)$ in the second step to get a cleaner expression.

For the SE kernel, recall from Theorem 3 and Corollary 1 that $\beta_m = \mathcal{O}(\sqrt{\widetilde{\gamma}_{m-1} + \log(1/\delta)}) = \mathcal{O}(\sqrt{(\log T)^{d+1} + \log(1/\delta)})$. Also recall that Theorem 2 tells us that $m \leq \overline{m} = \mathcal{O}((\log T)^{d+1})$. Therefore, for the linear kernel,

$$R_T = \mathcal{O}\left( (\log T)^{d+1} \sqrt{(\log T)^{d+1} + \log(1/\delta)} \log \frac{(\log T)^{d+1}}{\delta} \right)$$

$$= \mathcal{O}\left( (\log T)^{3(d+1)/2} \log((\log T)^{d+1}) \right), \tag{49}$$

in which we have again ignored the dependency on $\log(1/\delta)$ to obtain a cleaner expression.

The error probability of $\delta$ results from (a) conditioning on the event that $|f(x_s) - y_s| \leq \epsilon_s, \forall s = 1, \ldots, m$ which holds with probability of at least $1 - \delta/2$, and (b) Theorem 3 which also holds with probability of at least $1 - \delta/2$.

# G   Proof of Theorem 5

Consistent with the analysis of Q-LinUCB [38], in order to apply the theoretical guarantee provided by QMC (Lemma 1) for noise with bounded variance, here we need to assume that the noise variance is not too small: We assume that $\sigma > 1/3$. We have chosen the value of $1/3$ just for convenience, in fact, any $\sigma$ larger than $1/4$ can be used in our analysis here because these different values will only alter the constants in our regret upper bound which are absorbed by the $\mathcal{O}$.

Note that $\epsilon_s = \widetilde{\sigma}_{s-1}(x_s)/\sqrt{\lambda} \leq 1$, which follows because $\widetilde{\sigma}_{s-1}^2(x_s) \leq 1$ (which can be easily seen from (3) and our assumption that $k(x, x') \leq 1$) and $\lambda > 1$. As a result of the assumption of $\sigma > 1/3$, we have that $\sigma > 1/3 > 2^{\sqrt{2}}/8$. This, combined with $\epsilon_s \leq 1$, allows us to show that $\log_2^{3/2}\left(\frac{8\sigma}{\epsilon_s}\right) \geq 2^{3/4}$ and that $\log_2(\log_2 \frac{8\sigma}{\epsilon_s}) \geq 1/2$.

Next, we derive an upper bound on $\sum_{s=1}^{m} \frac{1}{\epsilon_s^2}$ which is analogous to (8). Note that in the case of a noise with bounded variance, the number of queries to the quantum oracle in stage $s$ is given by $N_{\epsilon_s} = \frac{C_2 \sigma}{\epsilon_s} \log_2^{3/2} \left( \frac{8\sigma}{\epsilon_s} \right) \log_2(\log_2 \frac{8\sigma}{\epsilon_s}) \log \frac{2\overline{m}}{\delta}$ with $C_2 > 1$. As a result, we have the following inequality which is in a similar spirit to (7)

$$\sum_{s=1}^{m} \frac{C_2 \sigma}{\epsilon_s} \log_2^{3/2} \left( \frac{8\sigma}{\epsilon_s} \right) \log_2(\log_2 \frac{8\sigma}{\epsilon_s}) \log \frac{2\overline{m}}{\delta} \geq \sum_{s=1}^{m} \frac{1}{3} \frac{1}{\epsilon_s} 2^{3/4} 2^{-1} \geq \frac{1}{3 \times 2^{1/4}} \sqrt{\sum_{s=1}^{m} \frac{1}{\epsilon_s^2}} \quad (50)$$

Now suppose that $\sum_{s=1}^{m} \frac{1}{\epsilon_s^2} > 9\sqrt{2}T^2$, then we have that $\sum_{s=1}^{m} N_{\epsilon_s} > T$ which is a contradiction. Therefore, we have that

$$\sum_{s=1}^{m} \frac{1}{\epsilon_s^2} \leq 9\sqrt{2}T^2. \quad (51)$$

Next, note that Theorem 1 is unaffected since its proof does not depend on the noise. Moreover, Corollary 1 also stays the same because it has only made use of (8) which gives an upper bound on $\sum_{s=1}^{m} 1/\epsilon_s^2$, and compared to (8), its counterpart (51) in the analysis here only introduces an extra constant factor which is absorbed by the bit $\mathcal{O}$ notation. Similarly, Theorem 2 is also unaltered for a similar reason, i.e., only an extra constant factor of $3 \times 2^{1/4}$ will be introduced to the right hand side of (30) and (34), which is absorbed by the big $\mathcal{O}$ notation.

Theorem 3 is also unaltered since its proof does not depend on the type of noise. In particular, the key underlying reason why it is unaffected is because for both types of noise, given a particular $\epsilon_s$, we choose the corresponding number $N_{\epsilon_s}$ of queries (to the quantum oracle) depending on the type of noise so that the error guarantee of $|y_s - f(x_s)| \leq \epsilon_s$ is satisfied. This allows us to make use of the self-normalizing bound from [9] for 1-sub-Gaussian noise.

Lastly, we need to modify the proof of Theorem 4. To begin with, the total regret from stage $s$ can now be bounded as

$$\frac{C_2 \sigma}{\epsilon_s} \log_2^{3/2} \left( \frac{8\sigma}{\epsilon_s} \right) \log_2(\log_2 \frac{8\sigma}{\epsilon_s}) \log \frac{2\overline{m}}{\delta} \times 2\beta_s \epsilon_s \sqrt{\lambda} = \mathcal{O}\left( \sigma \log_2^{3/2}(\frac{\sigma}{\epsilon_s}) \log_2(\log_2 \frac{\sigma}{\epsilon_s}) \log \frac{\overline{m}}{\delta} \beta_s \right). \quad (52)$$

Note that every $1/\epsilon_s$ is upper-bounded by $1/\epsilon_s \leq 3 \times 2^{1/4}T = \mathcal{O}(T)$ which can be easily inferred using (51). So, the total regret can be upper-bounded as

$$R_T = \mathcal{O}\left( \sum_{s=1}^{m} \sigma \log_2^{3/2}(\frac{\sigma}{\epsilon_s}) \log_2(\log_2 \frac{\sigma}{\epsilon_s}) \log \frac{\overline{m}}{\delta} \beta_s \right)$$

$$= \mathcal{O}\left( m\beta_m \log \frac{\overline{m}}{\delta} \sigma \log_2^{3/2}(\sigma T) \log_2 \left( \log_2(\sigma T) \right) \right). \quad (53)$$

As a result, for the linear kernel for which $m \leq \overline{m} = \mathcal{O}(d \log T)$ (Theorem 2) and $\beta_m = \sqrt{\widetilde{\gamma}_{m-1} + \log(1/\delta)} = \mathcal{O}(\sqrt{d \log T + \log(1/\delta)})$ (Theorem 3 and Corollary 1),

$$R_T = \mathcal{O}\left( d \log T \sqrt{d \log T + \log(1/\delta)} \log \frac{d \log T}{\delta} \sigma \log_2^{3/2}(\sigma T) \log_2(\log_2(\sigma T)) \right)$$

$$= \mathcal{O}\left( \sigma d^{3/2} \log^{3/2} T \log(d \log T) \log_2^{3/2}(\sigma T) \log_2(\log_2(\sigma T)) \right), \quad (54)$$

in which we have ignored all $\log(1/\delta)$ factors for a cleaner expression.

For the SE kernel for which $m \leq \overline{m} = \mathcal{O}(\log^{d+1} T)$ (Theorem 2) and $\beta_m = \sqrt{\widetilde{\gamma}_{m-1} + \log(1/\delta)} = \mathcal{O}(\sqrt{\log^{d+1} T + \log(1/\delta)})$ (Theorem 3 and Corollary 1),

$$R_T = \mathcal{O}\left( \log^{d+1} T \sqrt{\log^{d+1} T + \log(1/\delta)} \log \frac{\log^{d+1} T}{\delta} \sigma \log_2^{3/2}(\sigma T) \log_2(\log_2 \sigma T) \right)$$

$$= \mathcal{O}\left( \sigma (\log T)^{3(d+1)/2} \log((\log T)^{d+1}) \log_2^{3/2}(\sigma T) \log_2(\log_2 \sigma T) \right), \quad (55)$$

in which we have again ignored all $\log(1/\delta)$ factors for a cleaner expression.

# H Proof of Theorem 6

Our proof here follows closely the proof of Theorem 2 from the work of [2]. For consistency, in our proof here, we follow the notations from [38] and [2].

Since we focus on linear bandits here, we assume that the reward function is linear with a groundtruth $d$-dimensional parameter vector $\theta^*$: $f(x) = x^\top\theta^*, \forall x$. Again following [2] and [38] as well as many other works on linear bandits, we assume that $\|\theta^*\|_2 \leq S$, and $\|x\|_2 \leq L, \forall x$. We use $X_s$ to denote $X_s \triangleq [x_1, \ldots, x_s]^\top$ which is an $s \times d$ matrix, and denote $Y_s \triangleq [y_1, \ldots, y_s]^\top$ which is an $s \times 1$-dimensional vector of observations. We use $\zeta_{1:s} \triangleq [\zeta_1, \ldots, \zeta_s]^\top$ to denote the $s \times 1$-dimensional vector of noise. These notations allow us to write $Y_s = X_s\theta^* + \zeta_{1:s}$. We denote $V_s \triangleq X_s^\top W_s X_s + \lambda I$. Following [38], we use $\widehat{\theta}_s$ to denote the MLE estimate of the parameter $\theta^*$ given the observations after the first $s$ stages: $\widehat{\theta}_s = V_s^{-1} X_s^\top W_s Y_s$. Given these notations, we firstly have that

$$
\begin{aligned}
\widehat{\theta}_s &= (X_s^\top W_s X_s + \lambda I)^{-1} X_s^\top W_s (X_s\theta^* + \zeta_{1:s}) \\
&= (X_s^\top W_s X_s + \lambda I)^{-1} X_s^\top W_s \zeta_{1:s} + (X_s^\top W_s X_s + \lambda I)^{-1}(X_s^\top W_s X_s + \lambda I)\theta^* \\
&\quad - \lambda(X_s^\top W_s X_s + \lambda I)^{-1}\theta^* \\
&= (X_s^\top W_s X_s + \lambda I)^{-1} X_s^\top W_s \zeta_{1:s} + \theta^* - \lambda(X_s^\top W_s X_s + \lambda I)^{-1}\theta^*.
\end{aligned}
\tag{56}
$$

Therefore,

$$
\begin{aligned}
x^\top\widehat{\theta}_s - x^\top\theta^* &= x^\top(X_s^\top W_s X_s + \lambda I)^{-1} X_s^\top W_s \zeta_{1:s} - \lambda x^\top(X_s^\top W_s X_s + \lambda I)^{-1}\theta^* \\
&= \langle x, X_s^\top W_s \zeta_{1:s}\rangle_{V_s^{-1}} - \lambda\langle x, \theta^*\rangle_{V_s^{-1}}.
\end{aligned}
\tag{57}
$$

This allows us to use the Cauchy-Schwarz inequality to show that

$$
\begin{aligned}
|x^\top\widehat{\theta}_s - x^\top\theta^*| &\leq \|x\|_{V_s^{-1}} \left(\left\|X_s^\top W_s \zeta_{1:s}\right\|_{V_s^{-1}} + \lambda\|\theta^*\|_{V_s^{-1}}\right) \\
&\leq \|x\|_{V_s^{-1}} \left(\left\|X_s^\top W_s \zeta_{1:s}\right\|_{V_s^{-1}} + \sqrt{\lambda}\|\theta^*\|_2\right).
\end{aligned}
\tag{58}
$$

The last inequality follows because $\|\theta^*\|_{V_s^{-1}}^2 \leq \frac{1}{\lambda_{\min}(V_s)}\|\theta^*\|_2^2 \leq \frac{1}{\lambda}\|\theta^*\|_2^2$. Note that $X_s^\top W_s \zeta_{1:s} = \sum_{k=1}^s \frac{1}{\epsilon_k^2}\zeta_k x_k$.

Next, we make use of the self-normalized concentration inequality from Theorem 1 of [2]. Specifically, we will use $\{\frac{\zeta_s}{\epsilon_s}\}_{s=1}^\infty$ as to replace the stochastic process $\{\zeta_s\}_{s=1}^\infty$ (i.e., representing the noise, $\{\eta_s\}_{s=1}^\infty$ according to the notations from [2]), and use $\{\frac{1}{\epsilon_s}x_s\}_{s=1}^\infty$ to replace the vector-valued stochastic process $\{x_s\}_{s=1}^\infty$ (i.e., representing the sequence of inputs). Now we explain why this is feasible.

Similar to the proof of Theorem 3, we again denote the observation noise as $\zeta_s$: $y_s = f(x_s) + \zeta_s$, and define $\mathcal{F}_s$ as the $\sigma$-algebra $\mathcal{F}_s = \sigma(x_1, x_2, \ldots, x_{s+1}, \zeta_1, \zeta_2, \ldots, \zeta_s)$. Similarly, an immediate consequence is that $x_s$ is $\mathcal{F}_{s-1}$-measurable, and $\zeta_s$ is $\mathcal{F}_s$-measurable. Following a similar analysis to that used in the proof of Theorem 3 allows us to show that $\epsilon_k$ is $\mathcal{F}_{k-1}$-measurable (i.e., $\epsilon_k$ can be predicted based on $\mathcal{F}_{k-1}$). Therefore, we have that $\frac{1}{\epsilon_k}x_k$ is $\mathcal{F}_{k-1}$-measurable (i.e., $\frac{1}{\epsilon_k}x_k$ can be predicted based on $\mathcal{F}_{k-1}$) and that $\frac{\zeta_k}{\epsilon_k}$ is $\mathcal{F}_k$-measurable. Moreover, conditioned on $\mathcal{F}_{k-1}$, $\frac{\zeta_k}{\epsilon_k}$ is 1-sub-Gaussian. This is because the QMC subroutine guarantees that $|y_k - f(x_k)| \leq \epsilon_k, \forall k = 1, \ldots, m$ (with high probability), which means that the absolute value of the noise $\zeta_k = y_k - f(x_k)$ is upper-bounded by $\epsilon_k$: $|\zeta_k| \leq \epsilon_k$. Therefore, conditioned on $\mathcal{F}_{k-1}$, $\frac{\zeta_k}{\epsilon_k}$ is 1-sub-Gaussian.

Given that these conditions are satisfied, we can apply the self-normalized concentration inequality from Theorem 1 of [2] to $\{\frac{\zeta_s}{\epsilon_s}\}_{s=1}^\infty$ and $\{\frac{1}{\epsilon_s}x_s\}_{s=1}^\infty$ with (following the notations from Theorem 1 of [2])

$$
\begin{aligned}
V_s &= V + \sum_{k=1}^s \left(\frac{1}{\epsilon_k}x_k\right)\left(\frac{1}{\epsilon_k}x_k^\top\right) = \lambda I + \sum_{k=1}^s \frac{1}{\epsilon_k^2}x_k x_k^\top, \\
S_s &= \sum_{k=1}^s \frac{\zeta_k}{\epsilon_k}\left(\frac{1}{\epsilon_k}x_k\right) = \sum_{k=1}^s \frac{1}{\epsilon_k^2}\zeta_k x_k,
\end{aligned}
\tag{59}
$$

in which we have used $V = \lambda I$. This allows us to show that

$$\|S_s\|_{V_s^{-1}} = \left\|X_s^\top W_s \zeta_{1:s}\right\|_{V_s^{-1}} \leq \sqrt{2\log\left(\frac{\det(V_s)^{1/2}\det(\lambda I)^{-1/2}}{\delta}\right)} \tag{60}$$

which holds with probability of at least $1 - \delta$. Next, again making use of the trace-determinant inequality from (16) and using the aforementioned assumption that $\|x\|_2 \leq L$, we can show that $\det(V_s) \leq (\lambda + \frac{L^2}{d}\sum_{k=1}^{s}\frac{1}{\epsilon_k^2})^d$. Also note that $\det(\lambda I) = \lambda^d$. As a result, we can further upper-bound the equation above by:

$$
\begin{aligned}
\left\|X_s^\top W_s \zeta_{1:s}\right\|_{V_s^{-1}} &\leq \sqrt{2\log\left(\sqrt{\frac{(\lambda + \frac{L^2}{d}\sum_{k=1}^{s}\frac{1}{\epsilon_k^2})^d}{\lambda^d}}\frac{1}{\delta}\right)} \\
&= \sqrt{2\log\left(\sqrt{\left(1 + \frac{L^2}{d\lambda}\sum_{k=1}^{s}\frac{1}{\epsilon_k^2}\right)^d}\frac{1}{\delta}\right)} \\
&\leq \sqrt{2d\log\frac{1 + \frac{L^2}{\lambda}\sum_{k=1}^{s}\frac{1}{\epsilon_k^2}}{\delta}},
\end{aligned}
\tag{61}
$$

in which we have used $d \geq 1$ in the last step to get a cleaner expression.

Therefore, we can re-write (58) as

$$|x^\top\widehat{\theta}_s - x^\top\theta^*| \leq \|x\|_{V_s^{-1}}\left(\sqrt{2d\log\frac{1 + \frac{L^2}{\lambda}\sum_{k=1}^{s}\frac{1}{\epsilon_k^2}}{\delta}} + \sqrt{\lambda}S\right), \qquad \forall x, s, \tag{62}$$

in which we have also used the above-mentioned assumption of $\|\theta^*\|_2 \leq S$. Now again following [2], we plug in $x = V_s(\widehat{\theta}_s - \theta^*)$, which gives us

$$
\begin{aligned}
\left\|\widehat{\theta}_s - \theta^*\right\|_{V_s}^2 &\leq \left\|V_s(\widehat{\theta}_s - \theta^*)\right\|_{V_s^{-1}}\left(\sqrt{2d\log\frac{1 + \frac{L^2}{\lambda}\sum_{k=1}^{s}\frac{1}{\epsilon_k^2}}{\delta}} + \sqrt{\lambda}S\right) \\
&= \left\|\widehat{\theta}_s - \theta^*\right\|_{V_s}\left(\sqrt{2d\log\frac{1 + \frac{L^2}{\lambda}\sum_{k=1}^{s}\frac{1}{\epsilon_k^2}}{\delta}} + \sqrt{\lambda}S\right).
\end{aligned}
\tag{63}
$$

Therefore, we have that

$$\left\|\widehat{\theta}_s - \theta^*\right\|_{V_s} \leq \left(\sqrt{2d\log\frac{1 + \frac{L^2}{\lambda}\sum_{k=1}^{s}\frac{1}{\epsilon_k^2}}{\delta}} + \sqrt{\lambda}S\right). \tag{64}$$

This completes the proof of Theorem 6.

Now we briefly show how the improved confidence ellipsoid from our Theorem 6 leads to an improved regret upper bound for Q-LinUCB. Here we follow the proof of Theorem 3 from [38], and hence defer the detailed explanations of some of the steps to the proof there. To begin with, we have that

$$
\begin{aligned}
f(x^*) - f(x_s) &= (x^* - x_s)^\top \theta^* \\
&\leq \epsilon_s\left(\left\|\widetilde{\theta}_s - \widehat{\theta}_{s-1}\right\|_{V_{s-1}} + \left\|\widehat{\theta}_{s-1} - \theta^*\right\|_{V_{s-1}}\right) \\
&\leq 2\epsilon_s \times \left(\sqrt{2d\log\frac{1 + L^2T^2/\lambda}{\delta}} + \sqrt{\lambda}S\right) \\
&= \mathcal{O}\left(\epsilon_s\sqrt{d\log T}\right).
\end{aligned}
\tag{65}
$$

Here for simplicity, we have upper-bounded $\sum_{k=1}^{s}(1/\epsilon_k^2)$ by $T^2$ (Lemma 2) in the second inequality, and have omitted the dependency on $\log(1/\delta)$.

Then the total regrets in stage $s$ can be upper-bounded by

$$\frac{C_1}{\epsilon_s}\log\frac{2\overline{m}}{\delta}\times\mathcal{O}\left(\epsilon_s\sqrt{d\log T}\right)=\mathcal{O}\left(\sqrt{d\log T}\log\frac{\overline{m}}{\delta}\right) \tag{66}$$

Plugging in our tighter confidence ellipsoid (Theorem 6) into the regret analysis of QLinUCB, we have that the regret upper bound of QLinUCB can be analyzed as

$$\begin{aligned}
R_T &= \mathcal{O}(\sum_{s=1}^{m}\sqrt{d\log T}\log(\frac{\overline{m}}{\delta})) \\
&= \mathcal{O}(m\sqrt{d\log T}\log(\frac{\overline{m}}{\delta})) \\
&= \mathcal{O}(d\log T\sqrt{d\log(T)}\log(d\log T)) \\
&= \mathcal{O}\left(d^{3/2}(\log T)^{3/2}\log(d\log T)\right),
\end{aligned} \tag{67}$$

which gives the tighter regret upper bound for Q-LinUCB that we have discussed in the main text (Sec. 5.6).

## I  Regret Upper Bound for the Matérn Kernel (Proof of Theorem 7)

In this section, we modify our analysis to derive a regret upper bound for the Matérn kernel. We focus on bounded noise here (i.e., the first scenario in Lemma 1), since the analysis for noise with bounded variance (i.e., the second scenario in Lemma 1) is similar and follows from the analysis of App. G. In our analysis here, for simplicity, we ignore the logarithmic factors.

To begin with, note that Theorem 1 is not kernel-specific and hence still holds in our analysis here. It has been shown that for the Matérn kernel, the upper bound on the standard maximum information gain is $\gamma_T = \widetilde{\mathcal{O}}(T^{\frac{d}{2\nu+d}})$ [35]. Therefore, similar to our Corollary 1, for the Matérn kernel, we can use Theorem 1 to show that

$$\widetilde{\gamma}_m = \widetilde{\mathcal{O}}\left(\left(\sum_{s=1}^{m}\frac{1}{\epsilon_s^2}\right)^{\frac{d}{2\nu+d}}\right)=\widetilde{\mathcal{O}}\left(T^{\frac{2d}{2\nu+d}}\right). \tag{68}$$

Next, we modify the proof of our Theorem 2 to derive the corresponding upper bound on the total number of stages for the Matérn kernel, i.e., we discuss here how we modify the proof of Theorem 2 in App. D. To begin with, (68) suggests that there exists a $C > 0$ such that $\widetilde{\gamma}_m \leq C\left(\sum_{s=1}^{m}\frac{1}{\epsilon_s^2}\right)^{\frac{d}{2\nu+d}}$. Different from our analysis for the linear and SE kernels in App. D (i.e., (27) and (32)), the absolute constant $C$ here also contains logarithmic factors. Similar to (27), we can show that

$$m\log 2 = \log\frac{\det(V_m)}{\det(V_0)}=2\widetilde{\gamma}_m\leq 2C\left(\sum_{s=1}^{m}\frac{1}{\epsilon_s^2}\right)^{\frac{d}{2\nu+d}}. \tag{69}$$

This naturally leads to

$$\sum_{s=1}^{m}\frac{1}{\epsilon_s^2}\geq\left(m\frac{\log 2}{2C}\right)^{\frac{2\nu+d}{d}}, \tag{70}$$

which is analogous to (28). Next, similar to (29), the total number of iterations can be analyzed as

$$\sum_{s=1}^{m}\frac{C_1}{\epsilon_s}\log(\frac{2\overline{m}}{\delta})\geq\sum_{s=1}^{m}\frac{1}{\epsilon_s}\geq\sqrt{\sum_{s=1}^{m}\frac{1}{\epsilon_s^2}}\geq\sqrt{\left(m\frac{\log 2}{2C}\right)^{\frac{2\nu+d}{d}}}. \tag{71}$$

Now suppose $m > T^{\frac{2d}{2\nu+d}} \frac{2C}{\log 2}$, then this implies that

$$\sqrt{\left(m\frac{\log 2}{2C}\right)^{\frac{2\nu+d}{d}}} > T. \tag{72}$$

This equation, combined with (71), suggests that $\sum_{s=1}^{m} \frac{C_1}{\epsilon_s} \log(\frac{2\overline{m}}{\delta}) > T$, which is a contradiction. Therefore, we have that

$$m \le T^{\frac{2d}{2\nu+d}} \frac{2C}{\log 2} = \widetilde{\mathcal{O}}\left(T^{\frac{2d}{2\nu+d}}\right). \tag{73}$$

Next, note that our confidence ellipsoid in Theorem 3 does not depend on the choice of the kernel and hence also holds for the Matérn kernel.

Lastly, we can modify the proof of Theorem 4 (App. F). Specifically, we can adopt the result from (47): $R_T \le \mathcal{O}\left(m\beta_m \log \frac{m}{\delta}\right)$. For the Matérn kernel, Theorem 3 and (68) allow us to show that $\beta_m = \widetilde{\mathcal{O}}(\sqrt{\widetilde{\gamma}_{m-1}}) = \widetilde{\mathcal{O}}(\sqrt{T^{\frac{2d}{2\nu+d}}}) = \widetilde{\mathcal{O}}(T^{\frac{d}{2\nu+d}})$. Combining this with (73) allows us to show that for the Matérn kernel,

$$R_T = \widetilde{\mathcal{O}}\left(T^{\frac{2d}{2\nu+d}}T^{\frac{d}{2\nu+d}}\right) = \widetilde{\mathcal{O}}\left(T^{\frac{3d}{2\nu+d}}\right). \tag{74}$$

This completes the proof of Theorem 7.

## J  More Experimental Details, and Experiment on Real Quantum Computer

In our experiments, when implementing both the classical GP-UCB and our Q-GP-UCB algorithms, we use random Fourier features (RFF) approximation to approximate the kernel: $k(x, x') \approx \widetilde{\phi}(x)^\top \widetilde{\phi}(x')$. As a result, we can implement the weighted GP posterior following (6) as discussed in App. A, in which the $M$-dimensional ($M < \infty$) random features $\widetilde{\phi}(\cdot)$ are used to replace the infinite-dimensional RKHS features $\phi(\cdot)$. We have used $M = 100$ random features in the synthetic experiment and $M = 200$ in the AutoML experiment. As we have mentioned in the main paper (Sec. 6), we have implemented the QMC algorithm from the recent work of [20] using the `Qiskit` python package. We have used 6 qubits for the Gaussian noise and 1 qubits for the Bernoulli noise. For each experiment, we perform multiple independent trials (10 trials for the synthetic experiment and 5 trials for the AutoML experiment) and have plotted the resulting mean and standard error as the solid lines and error bars in each figure, respectively. For our Q-GP-UCB, we choose $\beta_s = 1 + \log s, \forall s \ge 1$ following the order given by the theoretical value of $\beta_s$ (Theorem 3). For classical GP-UCB, we tried different approaches to setting $\beta_s$: $\beta_s = 1 + \log s$, $\beta_s = \sqrt{2}$ and $\beta_s = 1$, all of which are commonly adopted practices in BO; we found that $\beta_s = \sqrt{2}$ and $\beta_s = 1$ have led to the best performances for GP-UCB in, respectively, the synthetic and AutoML experiments.

For the synthetic experiment, some important experimental details have been discussed in Sec. 6. The reward function $f$ used in the synthetic experiment is sampled from a GP with the SE kernel with a length scale of $0.1$. In the AutoML experiment, an SVM is used for diabetes diagnosis. That is, we adopt the diabetes diagnosis dataset which can be found at `https://www.kaggle.com/uciml/pima-indians-diabetes-database`, which is under the CC0 license. The task involves using 8 features to predict whether the patient had diabetes or not, which constitutes a two-class classification problem. We use $70\%$ of the dataset as the training set and the remaining dataset as the validation set. We aim to tune two hyperparameters of the SVM: the penalty parameter and the RBF kernel parameter, both within the range of $[10^{-4}, 1]$. For simplicity, for each of the two hyperparameters, we discretize its domain into 5 points with equal spacing. This results in a domain with $|\mathcal{X}| = 25$. Here we adopt the SE kernel: $k(x, x') \triangleq \sigma_0^2 \exp(-\|x - x'\|_2^2/(2l^2))$ with $\sigma_0^2 = 0.5$. Since we have implemented our algorithms (both classical GP-UCB and our Q-GP-UCB) using RFF approximation as discussed above, we performed a grid search for the length scale $l$ within $\{0.1, 0.2, \ldots, 1\}$ for both GP-UCB and Q-GP-UCB when generating the random features. We found that $l = 1.0$ leads to the best performance for GP-UCB and $l = 0.2$ works the best for Q-GP-UCB. This is likely because for our Q-GP-UCB, the effective noise is much smaller (thanks to the accurate estimation by the QMC algorithm), which allows us to use a smaller length scale (which can model more complicated functions) to accurately model the reward function.

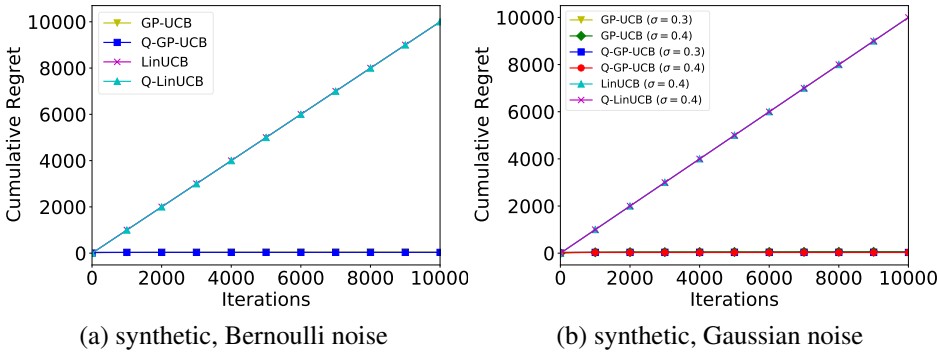

(a) synthetic, Bernoulli noise       (b) synthetic, Gaussian noise

Figure 2: Cumulative regret for the synthetic experiment with (a) Bernoulli noise and (b) Gaussian noise, with the additional comparisons with LinUCB and Q-LinUCB.

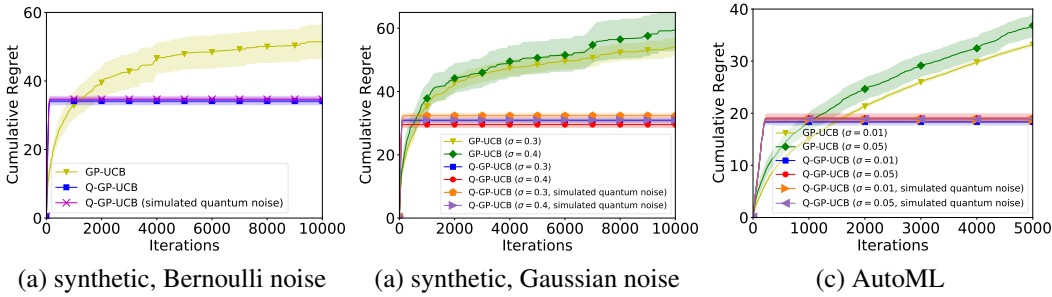

(a) synthetic, Bernoulli noise    (a) synthetic, Gaussian noise    (c) AutoML

Figure 3: Cumulative regret for (a) the synthetic experiment with Bernoulli noise, (b) the synthetic experiment with Gaussian noise, and (c) the AutoML experiment, with the additional comparisons with our Q-GP-UCB algorithm after considering quantum noise.

All our experiments are run on a computer with Intel(R) Xeon(R) Gold 6326 CPU @ 2.90GHz, with 64 CPUs. No GPUs are needed.

**More Experimental Results.** Fig. 2 additionally shows the cumulative regret for the LinUCB and Q-LinUCB algorithms in the synthetic experiment. Consistent with the results in Fig. 1 (d) in the main paper (Sec. 6) which has included the results for LinUCB and Q-LinUCB in the AutoML experiment, here LinUCB and QLinUCB again underperform significantly in Fig. 2 for both types of noises. These results further verify that in realistic experiments with non-linear reward functions, algorithms based on linear bandits are significantly outperformed by those based on BO/kernelized bandits. Fig. 3 additionally shows the results of our Q-LinUCB after considering quantum noise. As we have discussed in the main paper (Sec. 6), the results show that the quantum noise leads to slightly worse performances for our Q-GP-UCB, yet they are still able to significantly outperform classical GP-UCB.

**Results Using A Real Quantum Computer.** Here we additionally perform an experiment using a real quantum computer. Specifically, we ran our QMC subroutine on a real IBM quantum computer based on superconducting qubit technology (7 qubit, Falcon r5.11H processor) through the IBM Quantum cloud service. We ran our synthetic experiment with Gaussian noise on the system for only 1 trial (instead of 10 as in other simulations) due to time and resource constraints. The results in Fig. 4 show that although the performance of our Q-GP-UCB on real quantum computers is worse than those obtained without quantum noise and with simulated quantum noise, our Q-GP-UCB run on real quantum computers is still able to significantly outperform classical GP-UCB. This experiment provides a further demonstration for the potential of our Q-GP-UCB algorithm to lead to quantum speedup real BO applications.

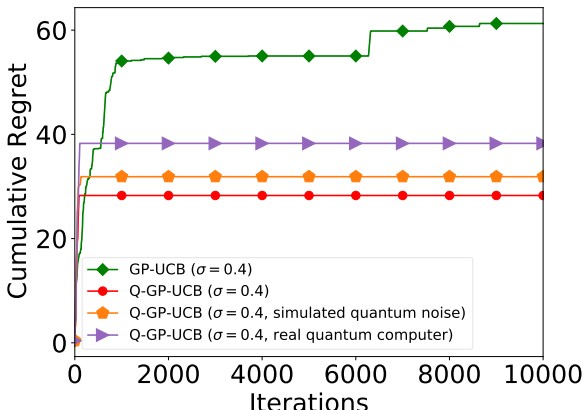

Figure 4: Results using **a real IBM quantum computer**. Cumulative regret for the synthetic experiment with Gaussian noise with $\sigma = 0.4$.

