# OpenReview forum: "Quantum Bayesian Optimization"
_NeurIPS.cc/2023/Conference — NeurIPS 2023 poster_

### Official Review · Reviewer_5kMD · 2023-07-04

**Soundness:** 3 good
**Presentation:** 4 excellent
**Contribution:** 3 good
**Rating:** 6
**Confidence:** 4

**Summary:**

The paper studies quantum kernelized bandits or Bayesian optimization (BO). Classically, in every iteration t=1,2,\ldots,T, a BO algorithm chooses an arm x_t and then queries the reward function f for a noisy observation y_t=f(x_t)+\zeta_t, where f can be non-linear and \zeta_t is a sub-Gaussian noise. The goal is to minimize the cumulative regret R_T=\sum_{t=1}^{T}[f(x^*)-f(x_t)]. In the quantum setting, every query to the reward function f at the arm x_t is replaced by a chance to access a quantum oracle or its inverse, which encodes the reward distribution for the arm x_t. Besides, a bounded noise or a noise with bounded variance are considered. The paper introduces the Q-GP-UCB algorithm which is the first BO algorithm able to achieve a regret bound of  O(poly log T), which is significantly smaller than the classical lower bound of \Omega(\sqrt{T}).

**Strengths:**

1. The paper provides the first quantum BO algorithm, which achieves a polylog(T) regret, beats the classical lower bound of \Omega(\sqrt{T}), and offers more evidence of quantum advantages over classical computers.

2. The result generalizes the previous quantum speedup for multi-armed bandits (MAB) and stochastic linear bandits (SLB) [32]. Besides, the paper improves the regret bound of SLB in [32] by improving the analysis of the confidence ellipsoid.

2. The paper is overall structured very well. The ingredients of the analysis for the Q-GP-UCB algorithm are listed in a readable way so I can quickly get the ideas behind it. I enjoy reading this paper very much.

**Weaknesses:**

1. I'm a bit doubtful about the technical contributions of this paper. Apparently, the basic framework of Q-GP-UCB follows from the weighted least squared estimator and the doubling trick in [32]. The key difference from [32] is the design of the weighted GP posterior distribution (see Eq. (3)), which looks very similar to the classical one (see Eq. (1)). Such a combination, of course, can be regarded as the main technical novelty, but the question is, I have no idea whether it raises inherent difficulties in the analysis.

**Questions:**

1. Please explain more about the technical difficulties in analyzing the Q-GP-UCB algorithm.

2. I notice that the authors of [32] showed their Q-LinUCB algorithm has a regret of O(log^{5/2} T) for bounded noise, but this paper said the regret is O(log^{3/2} T) (see line 350) when citing [32]. Is this because the analysis of [32] is not tight? Please confirm it since one of the contributions of this paper is an improvement over Q-LinUCB. If you indeed improve the regret from O(log^{5/2} T) to O(log^{3/2} T), this would increase the contributions of the paper.

**Limitations:**

Yes.

---

> ### Author Rebuttal · Authors · 2023-08-09
>
>
> We'd like to thank the reviewer for your insightful comments.
>
> ---
>
> > 1. I'm a bit doubtful about the technical contributions of this paper... Such a combination, of course, can be regarded as the main technical novelty, but the question is, I have no idea whether it raises inherent difficulties in the analysis.
>
> > 1. Please explain more about the technical difficulties in analyzing the Q-GP-UCB algorithm.
>
> The analyses of our algorithm indeed posed non-trivial difficulties. We have clarified and highlighted our technical novelty (including the difficulties) and contributions in our global response above (point 1). Here we give a brief summary:
> - The analysis of our weighted information gain (Theorem 1) is non-trivial, and is a novel contribution especially compared to quantum linear bandits [32] whose analysis didn't involve information gain. It may also be **of broader independent interest**.
> - The proof of our confidence ellipsoid (Theorem 3) also requires non-trivial techniques and insights, such as the recognition to apply the concentration inequality for 1-sub-Gaussian noise. It is also **novel because it is closely tied to our algorithmic design**.
> - We have used the techniques and insights from the proof of our Theorem 3 to **improve the confidence ellipsoid and hence the regret of quantum linear bandits [32]**, which is another important contribution of ours.
> - We've also made important empirical contributions by conducting a more realistic AutoML experiment and **an experiment using a real quantum computer**, both of which were not done by previous works on quantum bandits [32]. We also achieved significantly better performances than quantum linear bandits [32] and classical GP-UCB [28].
>
> In addition to the individual novel contributions listed above, another major aspect of our novelty and technical difficulty lies in **identifying the different required techniques** (e.g., weighted information gain, concentration for 1-sub-Gaussian noise) and **integrating them into a coherent analytical framework**, which we think is highly non-trivial.
>
> Therefore, we think that our work has made important novel technical contributions, which required overcoming non-trivial technical difficulties. Please refer to our global response above (point 1) for more details.
>
> ---
>
> > 2. I notice that the authors of [32] showed their Q-LinUCB algorithm has a regret of $\mathcal{O}(\log^{5/2}T)$ for bounded noise, but this paper said the regret is $\mathcal{O}(\log^{3/2}T)$ (see line 350) when citing [32]. Is this because the analysis of [32] is not tight? Please confirm it since one of the contributions of this paper is an improvement over Q-LinUCB. If you indeed improve the regret from $\mathcal{O}(\log^{5/2}T)$ to $\mathcal{O}(\log^{3/2}T)$, this would increase the contributions of the paper.
>
> If we understand correctly, you are referring to the upper bound on the **expected regret** of Q-LinUCB [32] (reported in Table 1 of [32]), which is indeed of the order $\mathcal{O}(\log^{5/2}T)$. To clarify, in our paper, we have only focused on **high-probability regret** (instead of expected regret), and the high-probability regret of Q-LinUCB [32] is of the order $\mathcal{O}(\log^{3/2}T)$ (Theorem 3 of [32], equation 37) which is consistent with what we have reported in our paper (line 350). The additional factor of $\mathcal{O}(\log T)$ in the expected regret results from the term $\log(\frac{m}{\delta})$ in the high-probability regret, because [32] has set $\delta=\frac{m}{T}$ to convert the high-probability regret to expected regret.
> In fact, the expected regret of our Q-GP-UCB with the linear kernel can also be easily derived, and it is also of the order $\mathcal{O}(\log^{5/2}T)$ which matches the expected regret of Q-LinUCB [32].
>
> More importantly, **for both high-probability regret and expected regret**, **our Q-GP-UCB (with the linear kernel) indeed improves over Q-LinUCB [32]** by a factor of $\mathcal{O}(\sqrt{d})$ (lines 348-350).
>
> ---
>
> Thank you again for your comments. We hope our additional clarifications could improve your opinion of our paper.

---

> > ### Comment · Reviewer_5kMD · 2023-08-12
> >
> > Thanks for your response. My doubts are well resolved.

---

> > > ### Author Response · Authors · 2023-08-15
> > > **Thank You for Your Response**
> > >
> > > Thank you for your response. We are glad to learn that your doubts are well resolved. We'll also add what we included in our response to the paper after revision to further improve our paper.
> > >
> > > Thanks again.

---

### Official Review · Reviewer_n5g2 · 2023-07-11

**Soundness:** 3 good
**Presentation:** 3 good
**Contribution:** 3 good
**Rating:** 7
**Confidence:** 3

**Summary:**

The paper studies the regret attainable for multi-armed bandits with non-linear reward functions when having access to a quantum oracle. For this setting they introduce the Quantum Gaussian Process Upper Confidence Bound (Q-GP-UCB) that with probability at least $\delta$ achieves regret:
- $\mathcal{O}((d\log{T})^{3/2}\log{d\log{T}})$ using the linear kernel,
-  $\mathcal{O}((\log{T})^{3/2\cdot(d+1)}\cdot (d+1) \cdot \log{\log{T}})$ using the squared exponential (SE) kernel,

when the noise is bounded and $d$ is the dimensionality of the input space. Similar rates are derived for noise with bounded variance.

This notable improvement over the classical fundamental limit is mostly attributed to the use of the Quantum Monte Carlo (QMC), and the improvement over the work of Wan et. al. 2022 (when instantiated with the linear kernel) is due to the paper's novel and tighter analysis of the confidence ellipsoid. They actually modify the proof of this prior work to attain the same rate as well.

Finally, they run experiments using the Qiskit package that demonstrate their algorithm's superiority over the classical variant, as well as the benefits of using the SE kernel in the more practical setting of AutoML.



**Strengths:**

- Intuitive and well explained use of staging to manage the growing number of samples fed to the QMC subroutine.
- Their weighing technique is very intuitive and by making the noise 1-sub-Gaussian allow them to use a self-normalized concentration inequality that improves the confidence ellipsoid and allows for better rates (also has nice intuitive meaning as enabling a measuring of weighted information gain).
- I particularly like that they use plug their improved analysis into the work of Wan et. al. to make their original algorithm match the rate the authors attain with the linear kernel. This is a nice way to both showcase the strengthened analysis, as well as to clarify to the community which things are key algorithmic behavior versus analysis artifacts.



**Weaknesses:**

- No core majorly novel algorithmic design / concept definition, i.e. results are due to elegant combinations of techniques and careful analyses rather than groundbreaking new concepts/approaches introduced.
- Minor typo: "encode" -> "encodes" (line 144).

**Questions:**

- Could you expand more on the empirical behavior of Q-GP-UCB in the initial stage? It is a bit hard to see given the large setting of total time-steps. For example, could you comment on when this may be an issue and when not?
- Do you have intuition for what would happen with other kernels being used?


**Limitations:**

- As per above, maybe more limitations regarding trade-offs of initial phase (and others) could be discussed in experimental section?

---

> ### Author Rebuttal · Authors · 2023-08-09
>
>
> We'd like to thank the reviewer for your valuable comments.
>
> ---
>
> > Could you expand more on the empirical behavior of Q-GP-UCB in the initial stage? It is a bit hard to see given the large setting of total time-steps. For example, could you comment on when this may be an issue and when not?
>
> The empirical behavior of our Q-GP-UCB in the initial stage (i.e., its regrets in the initial stage are relatively larger compared with the classical GP-UCB) is reasonable because our Q-GP-UCB explores a smaller number of unique arms than the classical GP-UCB initially (because we query every selected arm for multiple rounds). However, after the initial exploration, our Q-GP-UCB can quickly start performing reliable exploitation, because the accurate reward observations achieved thanks to our QMC subroutine allow us to quickly learn the reward function and hence find the optimal arm. Of note, the same behavior (i.e., relatively larger regrets in the initial stage) is also observed in [32] regarding the quantum linear bandit algorithm, which is similar to our algorithm using the linear kernel. So, we think that this behavior is not kernel-specific.
>
> Regarding your second question, we think this may be an issue when the problem is easy, i.e., when the reward function is easy to optimize (e.g., when the noise is too small or when the required number of iterations to find the optimal arm is too small). Specifically, when the problem is easy, classical BO algorithms such as GP-UCB may also be able to quickly find the optimal arm, which makes it difficult for the advantage of our Q-GP-UCB algorithm to manifest. On the other hand, this is unlikely to be an issue when the problem is relatively difficult, in which case we expect our Q-GP-UCB algorithm to consistently perform better, as we have demonstrated in our experiments.
>
> We'll follow you suggestion and add our discussions here to the experimental section.
>
> > Do you have intuition for what would happen with other kernels being used?
>
> As you suggested, we have additionally analyzed the regret of our algorithm for the Matern kernel, and included a detailed discussion in our global response above (point 2). Intuitively, for the Matern kernel, our regret bound also improves over the state-of-the-art regret in the classical setting when the reward function $f$ is sufficiently smooth.
> This in fact brings about an analogy between our work (as the first work on quantum BO) and classical GP-UCB [28]: GP-UCB also requires the function $f$ to be sufficiently smooth in order to achieve a sub-linear regret bound. Also similar to how GP-UCB was extended by later works, We leave it to future works to further improve our regret bound for the Matern kernel.
> We'll add the results and discussions about the Matern kernel (more details in the global response above, point 2) to the paper after revision, which we think will further improve the significance of our contributions.
>
> ---
>
> Thank you again for your feedback. We'll also follow your suggestion to correct the typo, and do a thorough check for other potential typos.

---

### Official Review · Reviewer_pGF9 · 2023-07-18

**Soundness:** 3 good
**Presentation:** 3 good
**Contribution:** 2 fair
**Rating:** 5
**Confidence:** 4

**Summary:**

This paper considers kernelized bandits also known as Bayesian optimization under a particular feedback model inspired by quantum computing. Under this model repeating sampling from the same point for N times reduces the noise to the level on $1/N$. That is significantly tighter than the classic setting where sampling the same point for N times reduces the noise to the level on $1/\sqrt{N}$. The paper introduces a UCB based algorithm, that is very similar to mini-META proposed in [6]. The proposed algorithm is referred to as Q-GP-UCB, and, in the case of SE kernel, obtains a polylogarithmic regret bound in the time horizon $T$ that is a significant improvement compared to $\sqrt{T}$ regret bound in the classic setting.

**Strengths:**

The problem is inspired by quantum computing and may be of broader interest. Overall this is an interesting formulation of kernel bandits.



**Weaknesses:**

The formulation and analytical techniques are similar to those of [32] in the case of linear bandits. That to some extent limits the novelty and contributions.

In terms of complexity, the regret bounds seem to scale as $\tilde{\gamma}_m^{1.5}$, where the number of unique points is bounded by $\tilde{\gamma}_m$ and an additional $\tilde{\gamma}_m^{0.5}$ is contributed by the regret on each unique point, due to confidence ellipsoid. Theorem 1 bounds this quantity by \gamma_{T^2}. For example, in the case of Matern kernels, that leads to a regret bound of $T^{(3d)/(4\nu+2d)}$ where $\nu$ is the smoothness of the kernel. This regret bound may be worse than the classic results of $T^{(\nu+d)/(2\nu+d)}$ when $d$ is large. I think this is an indication that the regret bounds presented in this work are sub-optimal and can be further improved. Given that there is no lower bound under this setting, that raises some doubts about the tightness of the analysis. The results should be seen as some initial attempt on the problem.

**Questions:**

- Considering the example above about Matern kenrel, could the authors comment on the tightness of the bounds? The results seem to be an improvement only in the case of SE kernel. Why they do not necessarily improve the regret bounds in the case of other kernels.

- If the standard mean and variance are used rather than the weighted ones, which step in the proof would fail? The number of unique points or the regret on each unique point?

**Limitations:**

Although the formulation seems interesting, the results do not seem tight. To some extent they follow the case of linear bandits. When it comes to more general kernels the regret bounds seem to even fail to be sublinear in some cases.

---

> ### Author Rebuttal · Authors · 2023-08-09
>
>
> We'd like to thank the reviewer for your constructive feedback.
>
> ---
>
> > The formulation and analytical techniques are similar to those of [32] in the case of linear bandits. That to some extent limits the novelty and contributions.
>
> We have clarified our technical novelty and contributions (especially compared with [32]) in our global response above (point 1). Here we give a brief summary:
> - The analysis of our weighted information gain (Theorem 1) is non-trivial, and is a novel contribution especially compared to quantum linear bandits [32] whose analysis didn't involve information gain. It may also be **of broader independent interest**.
> - The proof of our confidence ellipsoid (Theorem 3) also requires non-trivial techniques and insights, such as the recognition to apply the concentration inequality for 1-sub-Gaussian noise. It is also **novel because it is closely tied to our algorithmic design**.
> - We have used the techniques and insights from the proof of our Theorem 3 to **improve the confidence ellipsoid and hence the regret of quantum linear bandits [32]**, which is another important contribution of ours.
> - We've also made important empirical contributions by conducting a more realistic AutoML experiment and **an experiment using a real quantum computer**, both of which were not done by previous works on quantum bandits [32]. We also achieved significantly better performances than quantum linear bandits [32] and classical GP-UCB [28].
>
> In addition to the individual novel contributions listed above, another major aspect of our novelty and technical difficulty lies in **identifying the different required techniques** (e.g., weighted information gain, concentration for 1-sub-Gaussian noise) and **integrating them into a coherent analytical framework**, which we think is highly non-trivial.
>
> Therefore, we think our work has made important contributions especially compared with quantum linear bandits [32]. Please refer to our global response above (point 1) for more details.
>
> ---
>
> > ... For example, in the case of Matern kernels, that leads to a regret bound of $T^{(3d)/(4\nu+2d)}$ where $\nu$ is the smoothness of the kernel. This regret bound may be worse than the classic results of $T^{(\nu+d)/(2\nu+d)}$ when $d$ is large. I think this is an indication that the regret bounds presented in this work are sub-optimal and can be further improved. Given that there is no lower bound under this setting, that raises some doubts about the tightness of the analysis. The results should be seen as some initial attempt on the problem.
>
> > ...could the authors comment on the tightness of the bounds? The results seem to be an improvement only in the case of SE kernel. Why they do not necessarily improve the regret bounds in the case of other kernels.
>
> As you suggested, we've added a discussion of our regret bound for the Matern kernel in our global response above (point 2). Below we also give a (self-contained) discussion, with additional discussions on the tightness and improvement (over the classical setting) of our regret bound.
>
> For the Matern kernel, our regret bound of $\widetilde{\mathcal{O}}(T^{(3d)/(2\nu+d)})$ also improves over the state-of-the-art regret in the classical setting $\widetilde{\mathcal{O}}(T^{(\nu+d)/(2\nu+d)})$ **when the reward function $f$ is sufficiently smooth** (i.e., when $\nu > 2d$). So, even when $d$ is large, we still achieve an improvement if $\nu$ is large enough. We think this smoothness condition for the Matern kernel is reasonable, which can be justified by drawing analogy between our work (as the first work on quantum BO) and the classical GP-UCB [28]: For the Matern kernel, **the classical GP-UCB** [28] (with a regret bound of $\mathcal{O}(\gamma_T\sqrt{T})=\widetilde{\mathcal{O}}(T^{(\nu+3d/2)/(2\nu+d)})$) **also requires the function $f$ to be sufficiently smooth** (i.e., $\nu>d/2$) to achieve a sub-linear regret. This requirement for smooth functions was only removed in later works (e.g., [19,25,31]), which used sophisticated algorithmic designs and analyses to improve the regret of classical GP-UCB to $\mathcal{O}(\sqrt{\gamma_T T})=\widetilde{\mathcal{O}}(T^{(\nu+d)/(2\nu+d)})$. Therefore, we agree that for the Matern kernel, our regret bound may not be tight. However, given our discussions here, our regret bound for the Matern kernel is still a reasonable and important contribution (analogous to the regret of GP-UCB for the Matern kernel which also requires smooth functions), and we hope our work could also inspire future works to further improve the regret of quantum BO for the Matern kernel (similar to how the regret of GP-UCB was improved by later works).
>
> Lastly, we briefly summarize the contributions and **improvements** of our regret bound for different kernels: For the *SE kernel*, our regret bound significantly improve over the classical regret; for the *Matern kernel*, our regret bound improves over the classical regret when the reward function $f$ is sufficiently smooth; for the *linear kernel*, our regret bound improves over that of the quantum linear bandits [32].
>
> ---
>
> > If the standard mean and variance are used rather than the weighted ones, which step in the proof would fail? The number of unique points or the regret on each unique point?
>
> If the standard GP posterior mean and variance are used instead of the weighted ones, the proof of the number of unique points (i.e., the total number of stages, Sec. 5.2) would not be valid. Specifically, the proof of Equation (25) in Appendix D wouldn't go through, and hence we could no longer derive the upper bound on the total number of stages given in Theorem 2.
> Therefore, the weighted GP posterior regression (Sec. 4.1) is indispensable for deriving our theoretical results.
>
> ---
>
> Thank you again for your comments. We hope our additional clarifications could improve your evaluation of our paper.

---

> > ### Comment · Reviewer_pGF9 · 2023-08-10
> > **Suboptimality of the achieved regret bounds**
> >
> > Thanks for your response. In the case of Matern kernel, the regret bounds proven in this paper are in order of $\mathcal{O} (T^{3d/(2\nu+d)})$. This is worse than the optimal regret bound $\mathcal{O} (T^{(\nu+d)/(2\nu+d)})$ for standard kernel bandits when $\nu<2d$. Even when compared with the suboptimal $\mathcal{O} (T^{(\nu+3d/2)/(2\nu+d)})$ regret bound of GP-UCB, the regret bounds proven in this paper are worse when $\nu<1.5 d$. This is despite the observation that the noise concentrates faster, at a 1/N rate, in the quantum setting, in contrast to the $1/\sqrt{N}$ rate in the classic setting. It thus seems clear that the regret bounds proven in this paper are suboptimal in general. In addition, the reason for this suboptimality is not clear (where the difficulty comes from, what are the best regret bounds we hope for). I suggest the authors make this point clear in the paper to encourage future work on the topic.

---

> > > ### Author Response · Authors · 2023-08-11
> > > **Thank You for Your Reply**
> > >
> > > We agree that our regret bound for the Matern kernel implies that in general, our regret upper bound does not match the (unknown) lower bound and is hence not optimal. We'll revise the paper to make this clear.
> > >
> > > We'd like to add that the tightness of our regret upper bound is kernel-dependent, and we think that **our gap with the lower bound is much smaller for the SE kernel**. This is in fact also in a similar spirit to the classical GP-UCB [28]: its regret upper bound of $\mathcal{O}(\gamma_T\sqrt{T})$ is suboptimal for both the SE kernel and Matern kernel (compared with the known classical lower bound in [26]). However, for the classical GP-UCB, **the gap between the upper and lower bounds is much smaller for the SE kernel** (i.e., logarithmic gap) than for the Matern kernel (i.e., polynomial gap).
> > > Similarly, for our Q-GP-UCB, we agree that for the Matern kernel, there is likely a large gap (e.g., polynomial in $T$) between our upper bound and the (unknown) lower bound. However, we think that for the SE kernel, our gap is much smaller. This can also be supported by the fact that for the SE kernel, our regret upper bound is only $\mathcal{O}(\text{ploy}\log T)$, which is significantly smaller than the classical regret lower bound of $\Omega(\sqrt{T})$. Therefore, we think that our significantly improved regret bound for the SE kernel over the classical setting, which is our main contribution, is an important step forward for the community.
> > >
> > > Regarding the difficulty of achieving a tighter regret upper bound, we think the challenge lies in the need to come up with novel (likely more sophisticated) algorithmic designs. This is also analogous to the classical GP-UCB, whose regret bound was improved by later works via more sophisticated algorithmic designs and analyses (e.g., [19,25,31]). These works have improved the regret upper bound of the classical GP-UCB from $\mathcal{O}(\gamma_T\sqrt{T})$ to $\mathcal{O}(\sqrt{\gamma_T T})$, and it is interesting to explore whether the techniques they adopted can also be applied to our algorithm to attain an improvement similar to $\mathcal{O}(\sqrt{\gamma_T})$.
> > >
> > > As you have also suggested, we'll add the discussions here to the paper after revision, in the hope that our paper could inspire future works aimed at improving our regret upper bound especially for the Matern kernel.

---

### Official Review · Reviewer_Hp2A · 2023-07-27

**Soundness:** 3 good
**Presentation:** 3 good
**Contribution:** 2 fair
**Rating:** 5
**Confidence:** 4

**Summary:**

This paper studies Bayesian optimization with quantum reward oracles where the reward function $f$ lies in an RKHS space with the squared exponential kernel, and at every iteration after input is selected, we can access a quantum unitary oracle and its inverse that encode the noisy reward distribution. In such a setting, the authors introduce the quantum-Gaussian process-upper confidence bound (Q-GP-UCB) which achieves a regret upper bound of $\mathcal O(\text{poly log} T)$. To do it, they introduce a weighted GP regression and then analyze the growth rate of the weighted information gain. Next, they derive a tight confidence ellipsoid which gives a guarantee of the concentration of the reward function and the weighted GP posterior mean. They also show that their bound on the confidence ellipsoid improves that of the quantum linear UCB (Q-LinUCB) algorithm [32] over a factor $\sqrt{d}$, where $d$ is the input dimension. Finally, they show the performance of their proposed algorithm over the classical GP-UCB and Q-LinUCB, through a synthetic experiment and an experiment on automated machine learning

**Strengths:**

- The paper is well-organized and easy to read. The arguments, and comparisons with related works are clear and well-supported.
- The paper is the first to introduce the first quantum Bayesian optimization algorithm which enjoys a regret upper bound of $\mathcal O(\text{poly log} T)$.

**Weaknesses:**

However, I am concerned about the novelty of the used techniques in this paper. They seem to be an unsophisticated combination of common techniques from classical Bayesian optimization (e.g., see [9], [28], [30]), the classical bandits [9], and quantum bandits [32]. It follows that the regret upper bound of $\mathcal O(\text{poly log} T)$ for their proposed BO algorithm has the same order as that of [32] which is designed for linear bandits.

The authors claim that the recent quantum works on multi-armed or linear bandits are not able to solve sophisticated real-world problems with non-linear reward functions like their setting. However, from the technical point of view, Bayesian optimization under the setting that the reward function $f$ lies in an RKHS space is in fact a kind of linear bandit problem except that the input space is continuous.

Hence, I think that this paper is OK but not good enough from a technical point of view.

**Questions:**

- The authors seem not to consider the case of the Materm kernel. Is there any improvement in the regret bound with the aid of quantum computing in this case?
-  It would be interesting if the authors can provide a discussion on the lower bound of the regret bound of BO in the setting of quantum computing.

**Limitations:**

Yes

---

> ### Author Rebuttal · Authors · 2023-08-09
>
>
> We'd like to thank the reviewer for your insightful comments.
>
> ---
>
> > However, I am concerned about the novelty of the used techniques in this paper... the regret upper bound of $\mathcal{O}(\text{poly}\log T)$ for their proposed BO algorithm has the same order as that of [32] which is designed for linear bandits.
>
> We have clarified and highlighted our technical novelty and contributions (especially compared with [32]) in our global response above (point 1). Here we give a brief summary:
> - The analysis of our weighted information gain (Theorem 1) is non-trivial, and is a novel contribution especially compared to quantum linear bandits [32] whose analysis didn't involve information gain. It may also be **of broader independent interest**.
> - The proof of our confidence ellipsoid (Theorem 3) also requires non-trivial techniques and insights, such as the recognition to apply the concentration inequality for 1-sub-Gaussian noise. It is also **novel because it is closely tied to our algorithmic design**.
> - We have used the techniques and insights from the proof of our Theorem 3 to **improve the confidence ellipsoid and hence the regret of quantum linear bandits [32]**, which is another important contribution of ours.
> - We've also made important empirical contributions by conducting a more realistic AutoML experiment and **an experiment using a real quantum computer**, both of which were not done by previous works on quantum bandits [32]. We also achieved significantly better performances than quantum linear bandits [32] and classical GP-UCB [28].
>
> In addition to the individual novel contributions listed above, another major aspect of our novelty and technical difficulty lies in **identifying the different required techniques** (e.g., weighted information gain, concentration for 1-sub-Gaussian noise) and **integrating them into a coherent analytical framework**, which we think is highly non-trivial.
>
> Therefore, we think that despite achieving the same regret order of $\mathcal{O}(\text{poly}\log T)$ as quantum linear bandits [32], our work has overcome non-trivial additional challenges and made important novel technical contributions, which go beyond unsophisticated combination of common techniques. Please refer to our global response above (point 1) for more details.
>
> ---
>
> > The authors claim that the recent quantum works on multi-armed or linear bandits are not able to solve sophisticated real-world problems with non-linear reward functions like their setting. However, from the technical point of view, Bayesian optimization under the setting that the reward function $f$ lies in an RKHS space is in fact a kind of linear bandit problem except that the input space is continuous.
>
>
> We'd like to clarify that when the reward function $f$ lies in an RKHS space, it is a linear function w.r.t. the RKHS feature mapping $\phi(x)$, but **not a linear function w.r.t. the original input $x$**. In contrast, the quantum linear bandits [32] does assume that $f$ is a linear function w.r.t. the original input $x$. Therefore, our algorithm is able to model non-linear reward functions, which the quantum linear bandit algorithm [32] is incapable of. Thanks for pointing this out, we'll revise the paper to make this point clearer.
>
> ---
>
> > The authors seem not to consider the case of the Materm kernel. Is there any improvement in the regret bound with the aid of quantum computing in this case?
>
> As you suggested, we have additionally analyzed our algorithm for the Matern kernel (with smoothness parameter $\nu$), and included a detailed discussion in our global response above (point 2). Briefly, for the Matern kernel, our regret bound $\widetilde{\mathcal{O}}(T^{(3d)/(2\nu+d)})$ also improves over the state-of-the-art regret in the classical setting $\widetilde{\mathcal{O}}(T^{(\nu+d)/(2\nu+d)})$ when the reward function $f$ is sufficiently smooth (i.e., when $\nu > 2d$). We think that this smoothness condition is reasonable for our work (as the first work on quantum BO) because it is analogous to the classical GP-UCB [28]: GP-UCB also requires the function $f$ to be sufficiently smooth (i.e., $\nu>d/2$) in order to achieve a sub-linear regret bound. This requirement of GP-UCB for smooth functions (for the Matern kernel) was only removed by later works (e.g., [19,25,31]) which improved the regret bound of GP-UCB through sophisticated algorithmic designs and analyses; similarly, we also leave it to future works to further improve the regret bound of our quantum BO for the Matern kernel.
>
> Thank you for pointing this out. We'll add the results and discussions about the Matern kernel (more details in the global response above, point 2) to the paper after revision. We think that these additional results, given our already significantly improved regret for the SE kernel, will further improve the significance of our contributions.
>
> ---
>
> > It would be interesting if the authors can provide a discussion on the lower bound of the regret bound of BO in the setting of quantum computing.
>
> To the best of our knowledge, deriving a regret lower bound for quantum multi-armed bandits and quantum linear bandits is also still an open problem [32]. We agree that obtaining a regret lower bound in the quantum bandit setting is an interesting and important research problem, and we aim to explore this in future works.
>
> ---
>
> Thank you again for your feedback. We hope our additional clarifications could improve your opinion of our paper.

---

> > ### Comment · Reviewer_Hp2A · 2023-08-21
> > **Thanks for your response**
> >
> > Thanks to the authors to address my concerns, especially the regret bound for the Matern kernel. I have no further questions.

---

### Author Rebuttal · Authors · 2023-08-09


We'd like to sincerely thank all reviewers for your constructive feedback and for appreciating our contributions.
For example, Reviewer Hp2A and Reviewer 5kMD have acknowledged that our paper "introduces the first quantum BO algorithm", Reviewer pGF9 has commented that our work "may be of broader interest", Reviewer n5g2 "particularly likes" the way we "showcase our **strengthened analysis**" and acknowledges our "**elegant combinations of techniques and careful analyses**", Reviewer 5kMD has commented that our work "offers more evidence of quantum advantages over classical computers" and "**enjoys reading this paper very much**". We are deeply encouraged by these comments.

We have provided individual responses to your questions below, and have also highlighted two points here.

---

# 1. Technical novelty and contributions.

Here we clarify and highlight some of our major technical novelty and contributions:

- Weighted information gain (Theorem 1): Our proof of Theorem 1, despite following the analysis of [30], is non-trivial since it requires carefully tracking the impact of the weights $1/\epsilon_s^2$ throughout the analysis (lines 257-258). Our proof of weighted information gain, as well as its adoption in the analysis of quantum bandits, is also novel to the best of our knowledge, especially when compared with the analysis of quantum linear bandits [32] which did not involve information gain at all. More importantly, our Theorem 1 may be **of broader independent interest** for future works using weighted kernel ridge regression (Sec. 4.1).
- Confidence ellipsoid (Theorem 3): Despite following the analysis of [9], our proof of Theorem 3 also requires non-trivial analyses and insights, such as **the recognition to apply the concentration inequality for 1-sub-Gaussian noise**. To the best of our knowledge, our technical proof and insights here are **novel since they is closely tied to our algorithmic design** (specifically, our QMC subroutine which is required to guarantee 1-sub-Gaussian noise, lines 297-307), and they are the core reasons why we can improve over the regret bound of quantum linear bandits [32] (see below).
- Improvement over quantum linear bandits [32]: Our regret upper bound (when using the linear kernel) is tighter than the regret of quantum linear bandits [32] (Sec. 5.6). Additionally, by adopting our proof techniques and insights for our Theorem 3, we have **improved the confidence ellipsoid and hence the regret bound of quantum linear bandits [32]** (Theorem 6) to match our result. This is another important contribution of ours.
- Empirical contributions: Our empirical experiments also represent important contributions, as they may be significant steps towards assessing the real-world potential of quantum bandit algorithms. Specifically, to the best of our knowledge, our paper is the first work on quantum bandits to include a non-synthetic experiment (AutoML experiment, Fig. 1 c-d) and **an experiment using a real quantum computer** (Fig. 4 on page 27). Moreover, in our experiments, we've shown that our quantum BO significantly outperforms quantum linear bandits [32] and classical GP-UCB [28].


In addition to the individual novel contributions listed above, another major aspect of our novelty and technical difficulty lies in **identifying the different required techniques** (e.g., weighted information gain, concentration for 1-sub-Gaussian noise) and **integrating them into a coherent analytical framework**, which we think is highly non-trivial.

So, we think that our work has made important novel technical contributions compared with the previous works. We'll also revise our paper to further clarify our technical novelty and contributions.

---

# 2. Other kernels such as the Matern kernel.

In this work, we've focused on the commonly used squared exponential (SE) kernel, and significantly improved the regret bound over the classical setting. We've also analyzed the regret of our algorithm with the linear kernel, and shown that it achieves a better regret bound than quantum linear bandits [32].

For the Matern kernel with smoothness parameter $\nu$, we can in fact also derive a regret bound of the order $\widetilde{\mathcal{O}}(T^{(3d)/(2\nu+d)})$ (ignoring log factors, we'll add more details on this after revision). This **improves over the state-of-the-art classical regret upper bound of $\widetilde{\mathcal{O}}(T^{(\nu+d)/(2\nu+d)})$ when the reward function $f$ is sufficiently smooth** (when $\nu > 2d$). We think this smoothness condition for the Matern kernel is reasonable, which can be justified by drawing analogy between our work (as the first work on quantum BO) and the classical GP-UCB [28]: For the Matern kernel, **the classical GP-UCB** [28] (with a regret bound of $\mathcal{O}(\gamma_T\sqrt{T})=\widetilde{\mathcal{O}}(T^{(\nu+3d/2)/(2\nu+d)})$) **also requires the function $f$ to be sufficiently smooth** (i.e., $\nu>d/2$) to achieve a sub-linear regret. This requirement for smooth functions was only removed in later works (e.g., [19,25,31]), which used sophisticated algorithmic designs and analyses to improve the regret of classical GP-UCB to $\mathcal{O}(\sqrt{\gamma_T T})=\widetilde{\mathcal{O}}(T^{(\nu+d)/(2\nu+d)})$. Similarly, because our Q-GP-UCB is the first BO algorithm in the quantum setting (analogous to GP-UCB [28] in the classical setting), we think it's expected and reasonable that our Q-GP-UCB **also** requires the function $f$ to be smooth for the Matern kernel (in order to improve over the regret in the classical setting), and we **also** leave it to future works to further improve the regret bound of our Q-GP-UCB for the Matern kernel.

We'll add the discussions here (as well as the associated technical details) about the Matern kernel to the paper after revision. We think that these additional results, given our already significantly improved regret for the SE kernel, will further improve the significance of our contributions.

---

### Decision · Program_Chairs · 2023-09-21

**Decision:**

Accept (poster)

**Comment:**

The paper investigates a new version of (kernelized) multi-armed bandits with non-linear reward functions where the decision maker has access to a quantum oracle. With this setting the authors prove that we can go below the well-known \Theta(\sqrt(T)) regret lower bound. I particular, they introduce a new bandit algorithm called Quantum Gaussian Process Upper Confidence Bound (Q-GP-UCB) that with high probability can achieve O(polylog T) regret. This is an exciting result and definitely advances the state of the art of bandit theory and Bayesian optimization. It also opens new research questions (e.g., to what extent quantum oracles can enhance the bandit algorithms, etc).

The rebuttal phase was quite productive and the authors have clarified all the concerns of the reviewers. Given this, I strongly recommend accepting this paper.